



# The impact of elevation and flow dynamics on hydrological drought and wet spell characteristics in semi-arid southeast Arizona

Mengtian Lu[1,2], Pieter Hazenberg[2,3], Xiaohui Lei[4], and Hao Wang[4]

[1]College of Civil Engineering and Architecture,Zhejiang University, Hangzhou 310058, China.
[2]Department of Hydrology and Atmospheric Sciences, PO Box 210011, Tucson AZ 85721-0011, USA.
[3]Now at: Deltares, Postbus 177, 2600 MH Delft, The Netherlands.
[4]State Key Laboratory of Simulation and Regulation of Water Cycles in River Basins, China Institute of Water Resources and Hydropower Research, Beijing 100038, China.

**Correspondence:** Pieter Hazenberg (pieter.hazenberg@deltares.nl)

**Abstract.** Identification and understanding of the dominant control mechanisms of hydrological extremes has drawn worldwide attention in recent decades. However, detailed understanding of drought and wet spells within semi-arid regions has been hampered by the fact that identification is difficult for no flow conditions. Classification methods that have been developed for regions with perennial flow, do not work for ephemeral semi-arid rivers, while approaches for arid environments have difficulties to deal with seasonal runoff. Recently, a method was presented to identify hydrological extremes within semi-arid regions, by combining approaches developed for perennial flow and arid environments. However, this combined approach shows difficulties to identify drought and wet spells within semi-arid domains with a yearly precipitation cycle (e.g. monsoon). The current paper proposes to modify the combined method and make it suitable for these domains.

The modified combined approach presented here to identify hydrological extremes was applied to decade-long discharge observations from 12 different locations within the San Pedro basin in southeastern Arizona. These locations correspond to catchments covering multiple elevation bands and runoff characteristics. Southern Arizona receives the majority of its rainfall from the summertime North American Monsoon (NAM), with frontal systems providing additional precipitation in winter.

Using the modified method, the identified droughts and wet spells last longer compared to the previously defined combined procedure, and drought generally does not only start in spring at the end of the dry season. Furthermore, results show that if a drought or wet spell starts during the NAM or post-NAM season, it will generally last longer as compared to one that starts in winter or spring. This specifically holds for catchments with no perennial flow. By increasing the flow averaging interval, the new method also enables to observe multi-year drought and wet spells patterns. For the precipitation limited semi-arid San Pedro basin results show that multi-year wet spell and drought are rare. This is caused by the strong impact of the summertime NAM that generally acts both as a start and reset button for both types of hydrological extremes.





## 1 Introduction

Hydrological extremes strongly impact the landsurface as well as society. Drought is one of the most well-concerned natural disasters that can increase the frequency of wildfires, cause substantial damages to ecosystems and result in agricultural and economic crisis. For the United States, the average cost of drought is approximately $6–8 billion per year (Easterling et al., 2007; Dai, 2011). There are several types of drought, depending whether one focuses on the atmosphere, agricultural or hydrological systems (Wilhite and Glantz, 1985; Mishra and Singh, 2010; Pedro-Monzonís et al., 2015; Bachmair et al., 2016). Hydrological drought is defined as an evident shortage and decrease of surface and subsurface water resources, which harms water supply, riparian protection and other sectors of natural and human beings (Tallaksen and Van Lanen, 2004; Van Loon and Van Lanen, 2012). Opposite to drought, wet spells indicate wetter than average periods above a predefined threshold, resulting from excessive precipitation over a pronounced period of time. As the storage availability within a catchment is less during a wet spell, these periods have an increased risk for flooding (Zolina, 2014; Vinnarasi and Dhanya, 2016; Chaudhary et al., 2017; Ciric et al., 2017).

Considerable research has improved our understanding of hydrological drought and wet spell characteristics, focusing specifically on the assessment of individual documented cases (e.g., (Döll et al., 2003; Hannah et al., 2011; Van Lanen et al., 2013) as well as the impact of long-term climate variations (e.g. Andreadis et al., 2005; Sheffield and Wood, 2008a, b; Shukla and Wood, 2008; Sheffield et al., 2009, 2012; Dai, 2012; Hannaford et al., 2010; Prudhomme et al., 2011; Stahl et al., 2011; Corzo Perez et al., 2011; Wehner et al., 2011). Many of these studies used gridded land surface and hydrological model data and showed increased amounts of severe dry spells impacting the land surface in recent decades. This causes more extreme drought as climate gets warmer (Trenberth et al., 2003; Sun et al., 2007; Meehl et al., 2007; Dominguez et al., 2012; Zhao and Dai, 2015). Other studies have used statistical methods to reproduce the stochastic time series for dry and wet spell duration and intensity evaluation and improve our understanding (e.g. Akyuz et al., 2012; Borgomeo et al., 2015).

Research focusing on hydrological extremes using large-scale observational records have mainly focused on North America and Western Europe (e.g.,(Hisdal et al., 2001; Peel et al., 2005; Fleig et al., 2006; Stahl et al., 2010; Wilson et al., 2010; Van Lanen et al., 2013), as long-term observations from other regions are more limited (Van Lanen et al., 2013). These analyses predominantly used observations from humid and northern climates with perennial stream flow. Detailed understanding of hydrological extremes within semi-arid domains using distributed observations has received considerably less attention. Within these semi-arid regions, correct identification of hydrological drought is difficult due to the regular occurrence of zero flow conditions, making traditional identification approaches not suitable (Van Huijgevoort et al., 2012). For periods without precipitation and runoff, it is therefore difficult to distinguish between dry conditions, drought and extreme drought, which are of great importance to water resources management. As a result, many previous contributions excluded semi-arid domains from further analyses or instead made use of an incorrect higher threshold level (e.g. Corzo Perez et al., 2011).

To improve the identification of hydrological drought during no-flow conditions, Van Huijgevoort et al. (2012) developed an identification approach that combined the threshold level identification method, originally developed for perennial flow conditions, with the consecutive dry period method developed for arid domains. This combined procedure was able to identify





hydrological extremes specifically within transitional regions, where zero flow conditions are common but not the standard. However, as the current work will show, this method has difficulty correctly identifying the starting point of a drought and wet spell for regions with a strong seasonal precipitation signal (e.g. Monsoon). The focus of this paper is to gain an improved understanding of hydrological extremes within the semi-arid southwest US. A main benefit of focusing on this region is that considerable amounts of long-term observations are available. As landscape variability has been shown to have a control on

the hydrological response within this region (e.g. Fan et al., 2007), the current work will focus specifically on the impact of elevation and flow dynamics as dominant sources of control on the observed hydrological extremes.

For the southwest US increases in agricultural demand and a rapidly growing urban population have impacted this relatively water scarce region, affecting available streamflow and groundwater resources. Observations show a drying out of the southwest US during recent decades, with seasonal and inter-annual precipitation variability during the North American Monsoon (NAM)

becoming more extreme (Luong et al., 2017; Demaria et al., 2019). As urbanization and population are expected to further increase, this drying can potentially cause serious water availability challenges, with an expected increase in the frequency and severity of drought (Seager et al., 2007, 2015; Cayan et al., 2010; Garfin et al., 2013; Cook et al., 2015; Jones and Gutzler, 2016; Berardy and Chester, 2017). To limit the impact of human influences and increase our understanding on the variability of hydrological extremes within a semi-arid environment, the current paper focuses on the San Pedro river basin in southeast

Arizona. Within this basin limited human development has had relatively little impact on the natural system.

The San Pedro receives the majority of its precipitation during the summertime North American Monsoon as well as from wintertime frontal precipitation systems (Goodrich et al., 2008; Dominguez et al., 2012; Stillman et al., 2013). As a result, largest streamflow is observed in summer, with flow being considerable in winter at higher elevation. Long-term observations of river flow are available for multiple locations and elevations. As flow conditions are generally ephemeral, to identify drought

and wet spells from observed records, a combined identification procedure is necessary. Since, the combined procedure originally developed by Van Huijgevoort et al. (2012) has difficulty to identify the start of a given drought or wet spells for regions with a clear precipitation season, the current work proposes to adjust the procedure and make it applicable here.

This papers is setup as follows. Section 2 presents a detailed overview of the San Pedro basin and the observations used in this work. In Section 3 a detailed description of the combined identification method of Van Huijgevoort et al. (2012) is given,

followed by the adjustment proposed here to make it applicable for domains with a strong seasonal precipitation and runoff signal. The results presented in Section 4 focus on the overall variability of the dry and wet spells within the San Pedro basin (4.1), detailed statistics and probability analyses of event duration and start date (4.2), and a sensitivity analysis of the derived results (4.3). Last this work finished with a detailed discussion and conclusions (Section 5).

## 2 Study Region

### 2.1 Catchment Description

As explained in the Introduction, the current work deals with gaining an improved understanding on the occurrence and variability of hydrological extremes within the semi-arid San Pedro river basin in southeast Arizona (see Figure 1). The $12,200$





km$^2$ San Pedro River is one of few free-flowing perennial streams in southeast Arizona. Its headwaters lie in northern Mexico, from where it flows northward into Arizona, before merging with the Gila river near Hayden, Arizona, which is one of the main

tributaries of the Lower Colorado River. Several endangered species and migratory birds rely on the riparian forest situated along the main stem of the San Pedro river.

The lower elevations of the San Pedro basin receive about 300-350 mm of precipitation, while at higher elevations precipitation increases to over 500 mm. Of these amounts, about 2/3 originate from the North American Monsoon (NAM) when moisture from the Gulf of Mexico, Gulf of California, and Pacific Ocean is transported into the region, causing short-duration

and high-intensity rainfall from localized thunderstorms and mesoscale convective systems. Additional precipitation mainly occurs from winter time frontal systems carrying moisture from the Pacific Ocean and causes snow over the mountains and rain in the valleys (Goodrich et al., 2008).

### 2.2 Observations and streamflow characteristics

Streamflow characteristics generally follow precipitation, with the majority of flow occurring in form of short-duration pulses

during the NAM, quickly diminishing in size downslope due to infiltration losses into the ambient dry channel bed (Goodrich et al., 2004). At higher elevation, streamflow is perennial and receives an additional input from snowmelt in winter. As a result, the main stem of the San Pedro flows perennially until about St. David, Arizona (i.e. north of location 2 in Figure 1), after which it is dry the majority of the year.

Long-term observations from Charleston, Arizona (since 1912) show a significant decrease in streamflow during the last

century, as a result of urban development and agricultural practices (Pool and Goes, 1999). Figure 2 shows the yearly total accumulation for both the NAM and rest of the year. This figure shows a statistically significant decrease with about 1.12 mm decade$^{-1}$, corresponding to a sharp reduction over the 100 year period. However, this decrease is predominantly caused by a jump occurring in the late 1950's, with observed accumulations from before and after this jump not showing statistical significance.

In the current study, stream flow records from 12 different locations were used, representing different upstream areas, flow regimes and elevations (see Figure 1). These data were collected by the U.S. Geological Survey (USGS) (10 stations) and by the U.S. Department of Agriculture (USDA) within Walnut Gulch Experimental Watershed (WGEW) (2 stations), with half having over 50 years of observations (see Table 1). Stations 1–4 are situated along the main stem of the San Pedro River, stations 5, 6 and 10 are part of the Babocomari river basin, stations 7 and 8 are situated in the Huachuca Mountains and station

9 in the Mule Mountains, respectively. Last, stations 11 and 12 correspond to the Flumes 1 and 6 of WGEW. It should be noted that because of potential issues with the quality of daily discharge data from station 3, it was decided not to make use of the data observed before 1940.

To obtain a detailed understanding on the impact of flow behavior on drought and wet spells characteristics within the San Pedro, it was decided to divide the stations into three different categories according to their hydrological and geographical

characteristics: 1) No-Baseflow (NB) stations (1, 7, 11 and 12) that only observe flow from intense precipitation during summer and generally have no discharge the rest of the year, 2) Baseflow Upslope (BU) stations (6, 8–10) that have relatively small





**Table 1.** Overview of the runoff observations used in the current study, the period of daily observations available, size of the catchment upstream, and elevation of the measurement location. Different numbers correspond to those shown in Figure 1. Last column indicates the identified runoff regime as presented in Section 2.2. For the Location column, SP indicates the main stem of the San Pedro, B indicates the Babocamari river, while W indicates Walnut Gulch.

| Nr | Station Nr | Location | Period of Record | Years | Size (km$^2$) | Elevation (m) | Flow Regime |
|----|-----------|----------|------------------|-------|----------|---------------|-------------|
| 1 | USGS 09472050 | SP Redington Bridge | 1995-2017 | 23 | 8019 | 860 | NB |
| 2 | USGS 09471550 | SP Tombstone | 1967-2017 | 51 | 4507 | 1152 | BD |
| 3 | USGS 09471000 | SP Charleston | 1912-2017 | 106 | 3196 | 1205 | BD |
| 4 | USGS 09470500 | SP Palominas | 1930-2017 | 88 | 1909 | 1276 | BD |
| 5 | USGS 09471400 | B Tombstone | 2000-2017 | 18 | 793 | 1213 | BD |
| 6 | USGS 09471380 | B Huachuca City | 2000-2017 | 18 | 408 | 1676 | BU |
| 7 | USGS 09470700 | Banning Creek | 2001-2017 | 17 | 23 | 1453 | NB |
| 8 | USGS 09470800 | Garden Canyon | 1959-2017 | 59 | 22 | 1646 | BU |
| 9 | USGS 09470750 | Ramsey Canyon | 2000-2017 | 18 | 11 | 1684 | BU |
| 10 | USGS 09471310 | Huachuca Canyon | 2000-2017 | 23 | 11 | 1707 | BU |
| 11 | ARS WGEW Fl1 | W Tombstone | 1956-2017 | 61 | 149 | 1227 | NB |
| 12 | ARS WGEW Fl6 | W Tombstone | 1961-2017 | 56 | 95 | 1345 | NB |

upstream areas located at higher elevation upslope, and 3) Baseflow Downslope (BD) stations (2–5) with large upstream areas at lower elevation. As the name indicates, these last two categories generally have perennial flow conditions but can fall dry after prolonged periods without precipitation.

In Figure 3 the representative discharge characteristics for a single station in each runoff category are shown for three typical years, corresponding to a dry, normal and wet flow accumulation year, respectively. This figure clearly shows the differences among the three categories, with BU stations showing much higher runoff values as compared to the other two categories and NB stations having zero runoff for most of the year, except for summer when runoff is flashy. It can also be observed that for the upslope station (BU), additional runoff from snow melt occurs in winter, which is generally limited downslope (BD).

## 3    Identification of Hydrological Extremes

### 3.1    Combined Identification Method

A generic method to identify hydrological extremes within the transition zone climates (warm and dry), where zero runoff occurs regularly but not as often as in arid climates was developed by Van Huijgevoort et al. (2012). Classical hydrological drought and wet spell identification methods like the Standardized Runoff Index (e.g. Shukla and Wood, 2008; Wu et al., 2018) do not work well within these domains. Therefore, Van Huijgevoort et al. (2012) proposed to combine the threshold





level method (TLM) and the consecutive dry period method (CDPM) to identify hydrological extremes and more specifically drought.

The TLM is one of the most widely used approaches to identify hydrological extremes for rivers with perennial runoff (e.g. Yevjevich, 1967; Tallaksen et al., 1997, 2009; Fleig et al., 2006). For each given period of interest within a year (e.g. day,
month), the TLM calculates the corresponding discharge percentile, in comparison with the same period observed for other years. Periods with a percentile value below (e.g. $\leq 20\%$) or above ($\geq 80\%$) a predefined threshold are identified as in drought or wet spell, respectively. Unfortunately, for locations with zero flow conditions for a considerable amount of time throughout year (e.g. $> 20\%$), the TLM cannot distinguish between a dry period and a drought.

The consecutive dry period method (CDPM) was originally developed for arid regions, where precipitation and runoff events
are rare (e.g. Groisman and Knight, 2008; Deni and Jemain, 2009). For these environments, instead of identifying a drought on the bases of a given percentile for a given period, the CDPM counts the consecutive number of periods with zero discharge. On the basis of the cumulative distribution function (cdf), the consecutive zero-discharge lengths threshold corresponding to a predefined distribution percentile can be calculated (e.g. $\mathrm{cdf} \geq 0.8$). All periods with a consecutive zero discharge number higher or equal than the threshold value are identified as in drought.

Van Huijgevoort et al. (2012) combined the benefits of the TLM with the CDPM to identify hydrological extremes from daily land surface and hydrological model data, by taking the following steps:

1. For a given day of the year, apply the TLM to identify its corresponding discharge percentile in case no-flow conditions for that day occur less than $5\%$ of the time.

2. For all periods with zero flow, calculate the consecutive dry day number, followed by calculating the cumulative distribu-
tion function (cdf) of all zero-discharge period lengths according to the CDPM. Define the consecutive dry day threshold on the bases of this cdf.

3. Combine days that are in drought according to step 1, with those followed by zero discharge situations, calculating for each of these days the consecutive number of being either in drought or with no flow (dry). For those days with zero discharge, compare its consecutive drought/dry number to the cdf obtained in step 2 and derive the corresponding
percentile. As the highest values of the cdf correspond to the most extreme drought (low percentiles), the corresponding cdf value is substracted from one. Then, the final discharge percentile is found by rescaling this value by accounding for the fraction of time that discharge is positive for that given day.

4. For all days with both positive and zero flow, identify a drought and wet spell in case its corresponding discharge percentile is $\leq 20\%$ and $\geq 20\%$, respectively.

**3.2 Modifying the Combined Method**

Even though the combined method proposed by Van Huijgevoort et al. (2012) has been a major advancement to identify hydrological extremes within transition regions, the method has two potential issues. First, the final discharge percentile distribution





for a given day of the year does not have to be perfectly uniform between 0–100. This is the result of combining the original
TLM, where discharge percentiles are obtained for a given day, with the CDPM, which defines the cumulative density function

(cdf) on the bases of all dry days. Even though consecutive drought/dry numbers are rescaled (see step 3 of Section 3.1), this
does not guarantee a perfect uniform distribution between 0 and 100 for a given day.

Second, the combined method has problems within climate with a strong precipitation season (e.g. NAM). More specifically,
in case a drought starts during this season with positive discharge, it will generally stop identifying the drought at the moment
positive flow ceases. This is a direct consequence of the CDPM approach, which defines the cumulative density function on

the basis of the total consecutive length of each dry period. As a result, a small consecutive zero discharge day number will
almost always correspond to a much higher discharge percentile (unless it occurs right after a long-duration drought). For
southern Arizona, this can cause a drought starting during the NAM for low positive discharges, to cease in September as flow
ceases (as the consecutive drought/dry day number is small). If zero flow conditions continue to occur, the combined method
will identify the system as in drought again, later in the dry season somewhere in spring. The original combined method will

therefore identify two droughts (i.e. one in summer during the NAM and one in spring after a prolonged period without flow),
instead of a single continuous drought. An example of such a situation is given in Figure 4b for 2003–2004.

Given the issues associated with the combined method as described above, the current work proposes to modify the combined
method by adjusting steps 2 and 3 of the combined procedure described in Section 3.1. For step 2, instead of calculating the
cumulative density function (cdf) on the bases of the total length of each dry period, the distribution is defined for each day

of the year, with zero flow occurring $\geq 5\%$ of the time, using the observed consecutive zero flow numbers for the given day.
Then, in step 3, for a given observed consecutive drought/dry day number the corresponding discharge percentile is calculated
on the bases of the daily cdf.

Similar to Van Huijgevoort et al. (2012), in the current work a drought and wet spell were identified for a given day, in case
the discharge percentile is $\leq 20\%$ and $\geq 80\%$, respectively. Figure 4 shows the difference in identified discharge percentile

between the combined method as proposed by Van Huijgevoort et al. (2012) (old method) and the modified combined method
proposed here (new method). The new method enables identification of drought for days with a small consecutive drought/dry
flow number, which are generally observed at the end of the NAM season. Therefore, for these periods, less abrupt changes in
the identified discharge percentile are observed for the new method presented here (see also Fig. 4b). The fact that the observed
hydrological drought continues after precipitation ceases results in a considerable increase in the total number of drought days

for 2002–2005 and 2009–2011 ( Fig. 4c). On the contrary, for wetter years (e.g. 2000, 2006–2008), the new method will not
automatically identify a drought in late spring, when the consecutive drought/dry discharge number is relatively small. This
because it focuses on a given day of the year. For these wetter years, the new method considerably reduces the number of
drought days observed.

## 3.3   Drought and Wet Spell Identification

As indicated in the Introduction, many of the long-term discharge observations available are impacted by human influences,
which can cause both sudden shifts and continuous changes in the observed discharge series (Easterling and Peterson, 1995;



Easterling et al., 1996; Menne and Williams, 2005). It is necessary to identify and correct for these impacts to enable the identification of hydrological extremes using the new modified combined procedure proposed here. Therefore, for each station for each year, the mean daily value of all observations $\leq$ 25th percentile, between the 25th–75th percentile and those $\geq$ 75th

percentile were calculated. It was decided to calculate these three values as these correspond to low, average and high discharge observations, as human practices (e.g. building of dam, groundwater pumping, land use change) can impact different regimes of the hydrograph. For each of the yearly mean daily low, average and high discharge series, sudden changes or "shift points" were identified using the Pettitt test (Pettitt, 1979; Villarini and Smith, 2010). Statistically significant shift point years (p-value $< 5\%$) for each of the three series were used to cut the original observational data series into separate intervals, unless

sudden shift within the different series occur within less than 20 years from previous shifts or since the start of the timeseries. This ensures that each timeseries has a minimal length of 20 years. Next, for each of these timeseries without sudden shifts, continuous long-term changes in the yearly mean runoff data were identified using the Mann-Kendall test (Helsel and Hirsch, 1993; Kundzewicz and Robson, 2004). For statistically significant trends (p-value $< 5\%$) the observed runoff series were detrended. Both the Pettitt test as well as the Mann-Kendall test are commonly used to identify shift points and trends in

hydroclimatological timeseries (e.g. Yue et al., 2002; Villarini et al., 2009; Mallakpour and Villarini, 2016). For each detrended discharge interval series, the modified combined procedure as described in Section 3.2 was applied to identified hydrological extremes. Although both shifts and continuous changes were present in the discharge data observed within the the San Pedro, their occurrence overall was rare and only limited to a few locations.

As indicated above, a drought and wet spell were identified for each calender day in case the discharge percentile were below

or above the 20th and 80th percentile, respectively. These are further classified as a moderate, severe and extreme drought in case the percentile is 10–20%, 5–10% and <5%, respectively, and as a moderate, severe and extreme wet spell for the 80–90%, 90–95% and >95%, respectively. For each drought and wet spell the starting data and the total duration are calculated.

### 3.4   Sensitivity Analysis

The previous section presented the details on how to estimate hydrological extremes from long-term discharge observations

within the San Pedro river basin using the modified combined method. To limit the occurrence of short duration drought separated by short non-drought intervals and to assess the sensitivity of the implemented identification procedure, a number of pooling methods have been developed to correct for these short inter-droughts occurrences (e.g. Fleig et al., 2006; Van Loon and Van Lanen, 2012). Two common pooling procedures are the moving average procedure (MA-procedure) (e.g. Tallaksen et al., 1997) and the inter-event time method (IT-method) (Zelenhasić and Salvai, 1987).

The MA-procedure filters out the occurrence of short duration extremes and smooths observations by employing a moving average window, pooling mutually dependent droughts and wet spells. In line with previous work (e.g. Van Huijgevoort et al., 2012; Van Loon and Laaha, 2015), the MA-procedure was applied on the originally observed discharge data using 30-day left-sided moving average filter. The sensitivity of the MA-procedure was assessed by increasing the window size to 60, 90, 180, 365 days and 5 year, respectively. Increasing the MA-window allows to focus on longer-term variations, similarly to long-term





atmospheric drought where a one-year or multi-year PDSI has been used (e.g. Dai et al., 2004; Dai, 2011; Sheffield et al., 2012).

To assess the impact of pooling, the current work also used the IT-method by:

1. Calculate the inter-event time (days) between two identified droughts or wet spells. For an inter-drought period calculate the maximum percentile for the inter-drought period and for the inter-wet spell period the minimum percentile,

2. For two consecutive droughts, if inter-event time is less than a threshold $N$ and the inter-drought maximum percentile is less than Pu, both events and pooled and identified as one drought. Similarly, for two consecutive wet spells, if inter-event time less than a threshold $N$ and the inter-wet spell minimum percentile is less than Pl, both events and pooled and identified as one wet spell.

In the current work, it was decided to vary $N$ between 10, 30, 60 and 180 days, Pu between $30\%, 40\%, 50\%$, and Pl between 245 $70\%, 60\%, 50\%$, respectively. By allowing the maximum and minimum percentile for a inter-drought and inter-wet spell period to stay above and below $50\%$, it is ensured that the observed discharge always corresponds to a dry and wet period.

For the majority of the results presented here, usage was made of a 30-day MA window while no additional pooling using the IT-procedure was performed. However, to obtain an improved understanding on the sensitivity of the derived results, both the impact of a variable MA-window size as well as inter-event time characteristics were assessed.

## 4 Results

The modified combined method presented in the previous section was used to calculate the corresponding discharge percentile from daily observations within the San Pedro basin (see Section 3). These percentiles were used to identify the occurrence and duration of hydrological extremes, and their corresponding characteristics. The following sections present the results of these analyses, focusing on temporal variations (Section 4.1), the relationship between occurrence and duration (Section 4.2), and 255 the sensitivity of the derived results (Section 4.3).

### 4.1 Individual Basin

As indicated in Section 2.2, it was decided to group the 12 stations into three categories. For a representative station from each category, the observations, discharge percentiles and identified drought and wet spell are given in Figure 6 for the years 2002–2007. Even though these three locations correspond to different flow and upstream characteristics, each location experienced 260 drought from the summer to winter in 2002 and 2003 (see also Scott et al., 2004), and during the winter of 2005. Similarly, overlapping wet spells were observed during the summer of 2004, and summer and fall of 2006 and 2007. This shows the strong impact of the atmosphere, with dry and wet periods in precipitation dynamics strongly impacting the hydrological response throughout the basin.

Besides overlapping hydrological extremes, considerable variations are observed as well. For Station 11, where baseflow 265 is non-existent, Fig. 6a shows that when a drought or wet spell starts at the end of the NAM when precipitation ceases, the





hydrological response tends to stay locked in a given state, resulting in a longer-duration drought or wet spell that lasts until the next monsoon season (see the fall–spring period for the years 2002–2003 and 2004–2005 for Station 11).

For Stations 9 and 3 (Fig. 6b and c), where baseflow is an important source of moisture, the observed discharge percentiles vary more throughout the year. Furthermore, extreme hydrological drought conditions (<5%) were only experienced for rela-
tively short periods of time, as compared to Station 11 (Fig. 6a). Generally, drought and wet spells are more intermittent for these locations with baseflow. Even though the general drought and wet spell occurrences are similar between both stations, the upslope catchment (Station 9) experiences some more frequent short duration wet spells during both the NAM and winter. This can be related to the relatively small catchment area, with localized intense precipitation events having a large impact on the hydrological response. The impact of these localized events is much smaller within the larger catchment upstream of
Station 3.

To put these results in a longer-term perspective, Figure 7 shows the occurrence of a moderate, severe and extreme drought and wet spell for a given year and day. For Station 11 drought and wet spells generally last longer when starting at the end of the NAM and early fall (similar to what could be observed from Fig. 6). However, it can also be observed that when a hydrological extreme is observed during the NAM, its duration is shorter and terminates within the same season. As runoff mainly results
from high-intensity precipitation during the NAM and from tropical disturbances in fall, drought and wet spells are of relatively short duration when occurring during this season, indicating the strong impact of the atmosphere on the hydrological response. As precipitation and more specifically runoff is rare outside the NAM/fall season, the discharge response tends to stay locked within a given hydrological extreme state once precipitation-generating runoff ceases.

Drought and wet spells for the two locations with baseflow (Stations 9 and 3) are much more fragmented throughout time,
although longer duration drought and wet spells lasting multiple seasons can be observed (e.g. early 1980s). Both of these locations receive runoff from high-intensity precipitation during the NAM, from frontal precipitation and snow melt in winter as well as from shallow groundwater. Therefore, if a drought or wet spell starts at the end of the NAM or winter after precipitation/snow melt, there is a strong possibility that it is terminated during the following winter or NAM, respectively. As a result, the duration is shorter compared to stations without baseflow.

For both Stations 3 and 11 long-term discharge observations (>60 years) are available, which allows for assessment of long-term variations in drought and wet spells characteristics. The right column of Figure 7 shows the yearly fraction of drought and wet spell as well as the 5-year moving average. None of the three stations experienced a drought or wet spell that lasted the whole year (i.e. fractions are < 1). For Station 3 it can be observed that prolonged wet spells were observed during the 1950s and later 1980s/early 1990s, while the drought occurrences were most frequent during early 1960s, 1980s and 2000s. A similar
wet spell in the 1980s was also observed for Station 11, which also experienced more frequent drought during the early 1990s and 2000s. These latter drought have been related to increases in temperature over the last 25 years (e.g. Easterling et al., 2007; Woodhouse et al., 2010; Demaria et al., 2019), while some of the earlier peaks in hydrological extremes have been related to long-term variations in sea surface temperature (e.g. Ropelewski and Halpert, 1986; Sheppard et al., 2002; Ciancarelli et al., 2014).



## 4.2 All Station Categories

Figure 8 presents for each runoff category the starting day of the year and duration of a drought. It should be noted that drought durations < 30 days have been removed from the figure to focus only persistent hydrological extremes. Figure 8 shows strong differences in drought characteristics for basins with (BU and BD) and without (NB) baseflow. For the NB category from June–November (i.e. monsoon and post-monsoon season) the duration shows a bi-modal distribution, lasting either very short (<100 days) or much longer (>250 days). For this category, droughts generally start when positive discharge values cease early during the NAM or fall, and terminate once positive discharges from intense precipitation are observed again. As this generally happens either during the NAM or early fall (short duration) or during the subsequent NAM season in the following year (long duration), the distribution becomes bi-modal. Furthermore, all observed droughts for NB category stations within the San Pedro basin, terminate during the following NAM. This leads to a negatively sloping upper limit, where duration is inversely proportional with the time until the next monsoon (see Fig. 8a). Only one drought that started in winter was not terminated during the monsoon, but lasted an additional year until the following monsoon (i.e. duration > 400 days).

Due to the mechanisms identified above, the majority of observed droughts for the NB category, start during the summer or fall (i.e. monsoon and post-monsoon, see Fig. 8b), with fewer droughts starting in winter and spring (pre-monsoon). The bi-model behavior during the monsoon and post-monsoon season for the NB category, is also reflected in the boxplots in Fig. 8c, which show a large interquartile spread. This spread is much smaller for the winter and pre-monsoon, and its duration decreases for each season subsequently, as time until the next monsoon decreases.

The drought duration characteristics for stations with baseflow show much less variability throughout the year. For these locations, baseflow forms an important component of the runoff response (see also Figs. 3 and 6). Therefore, drought occurs relatively constant throughout the year (see Fig. 8b and c). However, similar to the NB category, for the non-monsoon seasons, a negatively sloping upper limit is observed, where drought duration is inversely proportional with the time until the next monsoon. This indicates that also for these basins, all droughts that start during the non-monsoon season, terminate once the next monsoon arrives. On the contrary, droughts starting during the monsoon for basins with baseflow, generally ceases in subsequent fall or winter.

In similar manner Figure 8d presents the start day and duration of observed wet spells for the three different categories. For basins without baseflow (NB), the probability of a wet spell is strongest during the monsoon and fall season, although a small secondary peak is observed during winter from rare flow events the result from frontal precipitation (see panel e). For the NB category, generally a strong difference in wet spells duration is observed between those starting during the monsoon and the rest of the year, where the former are of short duration (< 2 months) due to the flashy intermittent nature of runoff. On the contrary, wet spells occurring during the non-monsoon period vary considerably in duration, with maximum duration being inversely proportional to the time until the next monsoon. These large durations are the result of relative rare positive discharge values in combination with a low consecutive zero discharge day numbers (see Section 3.2). These latter conditions terminate during the next monsoon once positive flow is observed again.





For stations with baseflow, wet spell characteristics are different between the upslope (BU) and downslope (BD) stations. For the BD locations, even though wet spells start throughout the year, highest probability is observed during the monsoon, which subsequently decreases in the following seasons (see Fig 8b). This can be related to the strong impact of monsoon precipitation as the major source of moisture throughout the San Pedro. Furthermore, BD stations cover larger upstream areas and thus show more integrated signals that are less variable throughout time. Generally, the duration of a wet spell for BD stations is smaller compared to the other categories ($< 2$ months), indicating that even though baseflow forms an important moisture source, excesses diminish quickly over time in comparison with the BU stations. Contrary, the BU stations show a larger duration and a wider spread in the observed wet spell duration. This is expected to result from the potential stronger impact of shallow groundwater as a source of runoff for the upslope regions, which is slowly being released in subsequent months after a wet period (see also Fig. 3). Therefore, a given condition lasts longer. Furthermore, the upslope locations show a stronger bi-modal distribution in yearly wet spell occurrence, starting either during the monsoon or in winter (see Fig 8e). For both the BD and BU categories, the maximum duration shows again an upper limit with time until the next monsoon.

## 4.3 Sensitivity to Pooling and Averaging

The previous two sections showed considerable variations in drought and wet spell characteristics within the San Pedro basin. This section focuses on the sensitivity of these results to short duration drought and wet spells through pooling as well as increasing the the moving average window to smooth observations.

As indicated in Section 3.4, the IT-method used for pooling considers two parameters: 1) the upper/lower limit discharge percentile levels, and 2) the duration of the inter-event period. Figure 9 shows for different pooling percentile and inter-event duration thresholds, the drought and wet spell duration distribution. This figure shows that for the San Pedro basin, pooling has a clear impact on the number of drought and wet spells (see lower left corner of each panel), but has limited impact on the overall shape of the duration distribution. This indicates that for the San Pedro pooling predominantly reduces the number of short duration drought and wet spells and merges these with a given longer duration extreme. Overall, the biggest impact of pooling is observed for the maximum upper drought percentile threshold ($Pu = 50\%$)) (top right panel Fig. 9) or minimum lower wet spell percentile threshold ($Pl = 50\%$)) (bottom right panel Fig. 9). For these highest thresholds, increasing the inter-event time from 10 to 180 days, strongly decreases the occurrence of short duration drought and wet spells ($\leq 20$ days) and increases probabilities for durations $\geq 60$ days. This indicates that these shorter duration droughts have become part of a longer-duration extreme. Results also show that the probability for a long duration drought or wet spells ($\geq 400$ days) is very rare, with pooling having no considerable effect.

In Figure 10 the impact of increasing the moving average (MA) window from 30 days to 5 years is shown (see also Section 3.4). As can be observed from this figure, the distributions for both day of the year (left of each panel) as well as duration (right of each panel) show considerable variations for different MA windows. As window size increases, the number of identified hydrological extremes decreases. The impact of this is especially visible for short duration drought and wet spells ($\leq 30$ days), with a strong decrease of this first probability maximum as window increases (right of each panel). For longer averaging windows (Fig. 10e and f right) the impact of these short duration extremes reduces considerably. Instead, a maximum around





a one year duration can be observed, which corresponds with the occurrence of the NAM. It should be noted that even for one and five year MA windows, multi-year hydrological extremes are rare.

For the day of the year, all panels show a first maximum that occurs during the NAM, indicating its dominance as the climatic

driver for hydrological extremes. It can also be observed that for all averaging windows $\geq 60$ days, the first maximum of the wet spell distribution occurs about 2–4 weeks earlier than for a drought. This indicates that a wet spell can result from a few intense precipitation events at the beginning of the monsoon, while in order for a hydrological drought to establish during the NAM, a longer period with below average precipitation is needed.

For increasing backward-looking MA windows, the impact of summertime discharge is present longer. This causes the

second maximum of the occurrence distribution for both droughts and wet spells, which for a 30-day window to occur in September (Fig. 10a), to be postponed until October, December, and February for averaging windows of 60, 90 and 180 days, respectively. Many of these droughts and wet spells, last until the following monsoon, which impact the duration distribution (right figure of each panel). This second maximum does not occur when an averaging window of 1 or 5 years are used. Instead, for these windows this maximum becomes part of the first maximum observed during the NAM. For these largest two window

sizes, the first maximum shows a significant number of drought and wet spells that last until the next year or even the year after (i.e. one or two year duration).

A third maximum in the occurrence distribution can be observed during winter period (January–February). As stated before, frontal winter precipitation and snow melt form an important secondary moisture source (see also Fig. 3). Therefore, an excess or reduced amount of winter precipitation can cause a hydrological extreme to start in winter. Since these hydrological extremes

are not an artifact of the applied window size, a winter peak in the distribution is observed irrespective of window size. It should be noted although that the number of drought and wet spells occurring during this period is much less as compared to the NAM. The majority of drought and wet spells that start in winter last until the next monsoon (see also Section 4.2). Only for the largest MA windows (bottom panels) a few extremes that start in winter are not terminated in summer, indicating that for these few extremes the NAM was not able to cause a reset.

For a one-year MA window, Figure 11 shows the identified drought and wet spells and fitted distributions for day of the year and duration for the three different runoff categories. It can be observed that the NB and BD categories show a stronger overlap with the majority of extremes starting during the NAM followed by the winter period. Also, the majority of drought and wet spell starting in winter terminate during the following monsoon. The impact of the winter period is less dominant for the BU category, although it should be noted that the number of observed extremes for this category is less as for these stations no

data prior to 1995 is available. Besides during the NAM, hydrological extremes start more widely throughout the year for the BU category. As stated before, the uplands receive a considerable amount of precipitation outside the NAM season. Therefore, these regions stay more hydrologically active, resulting in drought and wet spells to can start throughout the year. However, hydrological extremes starting during the non-monsoon period generally terminate during the following monsoon.



## 5 Discussion

The current paper presents a detailed analysis on observed hydrological extremes within the semi-arid San Pedro river basin and the impact of elevation and flow dynamics. As no flow conditions are common at lower elevations in this region, the combined hydrological extreme identification method originally developed by Van Huijgevoort et al. (2012) was adjusted and made applicable for regions with a strong periodic precipitation signal (i.e. the NAM). The newly combined procedure still merges the TLM and CDPM. However, the runoff percentiles of a given drought/dry day are now defined in comparison with

observation from the same day for other years. A benefit of this adjustment is that a drought that occurs during positive flow does not terminate once no-flow conditions start.

This work shows considerable variations in the characteristics of hydrological regimes between the three different runoff categories indicating the impact of elevation and flow conditions. For the three runoff categories defined here, the correspondence changes when increasing the MA window from thirty days to one year. Using a thirty day MA window, Sections 4.1 and 4.2

show considerable overlap in the drought and wet spell characteristics for the runoff categories containing baseflow (BU and BD). These categories receive a considerable amount of runoff from winter precipitation and shallow groundwater upslope, and stay hydrologically active during the non-monsoon period. In comparison, NB category stations only flow regularly during the non-monsoon and post-monsoon period. Figure 12 shows that this results in shorter duration hydrological extremes for the BU and BD category. Since the upland regions (BD) generally are the source of runoff for lower elevations (BU), especially

for the non-monsoon period, similar characteristics in hydrological extremes for these categories are observed.

When increasing the MA window to one year the correspondence between the BU and BD runoff categories changes. Figure 13 shows the observed drought (red) and wet spells (blue) since 1968. Generally, a given drought or wet spell occurs simultaneously throughout the basin, indicating the strong control of climate on observed hydrological extremes. However, more variability can be observed for the BU category, with the BD category showing more correspondence with NB stations.

It is anticipated that these changes for the BD category with increasing the MA window results from the strong impact of the NAM. At higher elevation, precipitation generating runoff occurs more regularly throughout the year (see Fig. 12). For short duration averaging windows (i.e. 30 days) runoff pulses generated upslope impact flow conditions at the lower elevation. However, for yearly averages, these pulses have limited effect on total accumulations downslope, reducing the impact of non-monsoon runoff on the occurrence of hydrological extremes. Transmission losses potentially play an important role here

(Goodrich et al., 2004), with flow conditions becoming smaller with travel distance. This enhances the role of climate as the dominant control mechanism on longer time scale. As a result, at lower elevations the NAM is the dominant driver for both categories with (BD) and without (NB) baseflow. At these timescales, drought and wet spells starting during the NAM, either cease in subsequent months or last until the following monsoon (see also Fig. 11).

Contrary to changing the MA window size, pooling was shown to have limited impact in the current work. For the San Pedro

using a inter-event period length of 30 days results in the strongest reduction in short-term drought and wet spells. Although the overall number of observed hydrological extremes decreases with increasing inter-event duration, pooling mainly impacts the occurrence of longer-duration drought or wet spells (≥ 3 months). This indicates that the occurrence of multiple short duration





hydrological extremes (< 20 days) is rare, which can be related to the sparse character of precipitation as observed within the San Pedro (Stillman et al., 2013).

Groundwater has been shown to be of similar importance as climate in controlling hydrological extremes in temperate regions (Peters et al., 2003; Van Loon and Van Lanen, 2012; Van Lanen et al., 2013). The current work shows that at the scale of the San Pedro groundwater has no serious impact on observed hydrological extremes. However, it is anticipated that for the uplands, which receive a considerable amount of runoff from groundwater, the subsurface does have some secondary control. More specifically, Section 4.2 showed that groundwater within the upland regions more strongly affects wet spell

characteristics, while this behavior is less evident for droughts. Van Loon and Laaha (2015) speculated that for drier climates, catchment characteristics are dominant in determining hydrological drought duration, with spatial variation in hydrological extremes being more pronounced for catchments with large elevations difference. Figure 12 shows that this clearly holds for the San Pedro basin, where the three regions show a clear distinction in the average drought and wet spell duration. Although, its impact is secondary to climate.

Changes in climate have been shown to have a strong impact on the hydrological response and the occurrence of extremes. The southwest US has been drying out during recent decades, with seasonal and inter-annual precipitation variability during the North American Monsoon (NAM) becoming more extreme (Prein et al., 2015; Solander et al., 2017; Luong et al., 2017). This also holds for the San Pedro, with increasing temperatures and an increased number of extreme precipitation events (Demaria et al., 2019). Section 2.2 explained that long-term observations available for the San Pedro, show no statistically significant

changes in measured runoff after 1960. Furthermore, Figure 7 (right panels) shows no significant changes in the occurrence of drought and wet spells for the same period. Based on a theoretical analysis, Botter et al. (2013) hypothesized that for basins where the mean inter-arrival between flow-producing rainfall events is larger than the typical duration of flow pulses (as for the San Pedro), a wider range in stream flow is observed between events. These basins with erratic hydrological regimes were hypothesized to have a stronger resilience to climate fluctuations. This indeed holds for the San Pedro where observed

hydrological extremes show resilience to changes in climate. Even though temperature has increased and locally precipitation has become more intense (Demaria et al., 2019), their impact on short-duration flow events at the catchment scale is relatively small and results in no significant changes in observed drought or wet spell characteristics (see also Fig. 7).

## 6 Conclusions

This paper focuses on the variability of observed drought and wet spells within the semi-arid San Pedro basin in southeast
Arizona and the impact of elevation and flow dynamics. Correct identification of hydrological extremes in semi-arid climates and more specifically drought, has been complicated due to intermittent flow conditions. Van Huijgevoort et al. (2012) improved the identification of hydrological extremes for these semi-arid domains by combining the threshold level method for temperate domains with the consecutive dry period method for arid domains. As this combined method has difficulties to correctly identify drought for regions with a strong precipitation season, like the North American Monsoon (NAM) for the San Pedro,
the current work proposes to improve this procedure by evaluating a given dry day with respect to other years. This ensures

that the percentile distribution for each day becomes correctly uniform and that a drought continues to persist when moving from positive to zero flow conditions. In comparison to the original combined method, the approach presented here results in fewer short-duration drought and instead increases their overall duration.

Using the modified combined method to identify hydrological extremes, this work shows that for the semi-arid San Pedro basin drought and wet spells can generally be classified into three main groups:

1. Short duration hydrological extremes that occur during the NAM/early fall

2. Hydrological extremes starting in winter and lasting until the next monsoon

3. Long-duration hydrological extreme that start during the NAM/early fall and last until the following NAM

These groups clearly show the strong impact of the NAM on the occurrence and duration of hydrological extremes as the summer/early fall is the dominant provider of moisture to the region. Because of this strong impact, multi-year hydrological extremes are generally rare even when applying a one-year averaging windows. For the San Pedro, the NAM therefore acts as a start button, which potentially continues until the non-monsoon period but almost definitely will terminate during the following NAM. This results in a hydrological response that shows resilience in observed hydrological extremes from observed changes in climate.

Some variations in hydrological extremes can be observed throughout the San Pedro (see e.g. Fig. 12). For locations with baseflow, drought and wet spell are more variability throughout the year and shorter in duration as compared to locations without baseflow. This especially holds for the upland domains where more regular precipitation and the occurrence of a shallow groundwater reservoir, results in more variable flow conditions during the non-monsoon periods.

*Code availability.* The numerical code developed in the R-language to identify hydrological extremes is available up request by contacting the authors.

*Data availability.* The runoff data can be obtained from https://waterwatch.usgs.gov/ for locations 1–10 and for locations 11–12 at https://www.tucson.ars.ag.

*Author contributions.* M. Lu performed analyses presented herein as designed by P. Hazenberg. X. Lei and H. Wang were involved in the supervising progress and contributed to the final manuscript.

*Competing interests.* No competing interests are present.



*Acknowledgements.* We thank the group of dedicated staff and technicians at USGS in Arizona and at USDA-ARS Walnut Gulch Experimental Watershed that have professionally maintained the network of runoff gauges used in this study. M. Lu would like to acknowlegde the support from the China Scholarship Council (CSC) enabling her to perform research in the US. This paper was jointly supported by National Key R&D Program of China (2017YFC0406004), the National Natural Science Fund (51709273) and the Key R&D Program of Power Construction Corporation of China (DJ-ZDZX-2016-02).



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





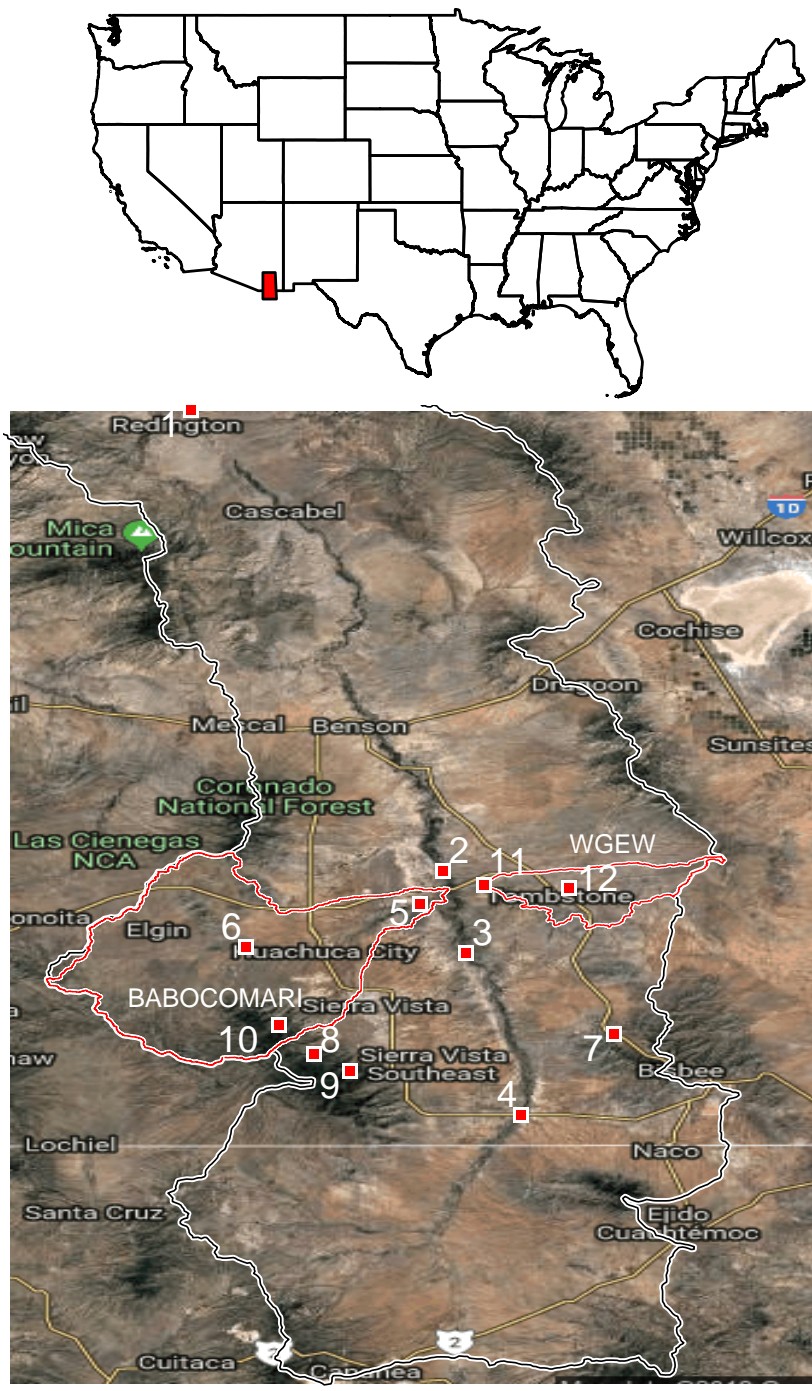

**Figure 1.** Location of San Pedro river basin in southeast Arizona and the location of the 12 runoff gauges used in this work. Background map was created using Rgooglemaps library (Loecher and Ropkins, 2015) in R (R Development Core Team, 2008) using ©Google. (n.d.).



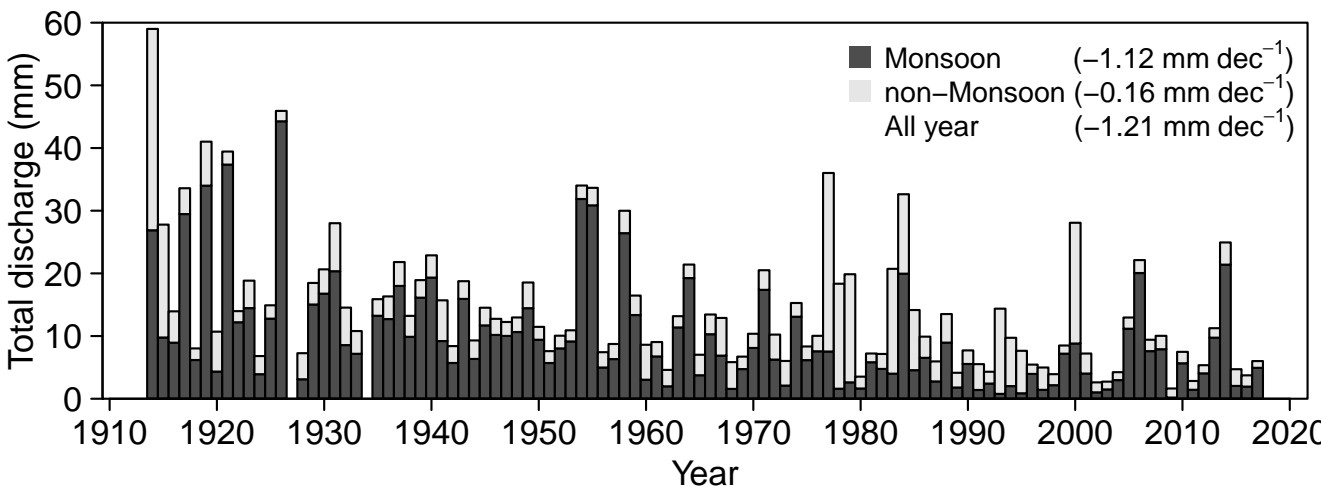

**Figure 2.** Total yearly runoff accumulation as observed at Charleston, Arizona (Location 2 in Fig. 1) during both the North American Monsoon and rest of the year. Top-right values in parenthesis indicate the average change per decade as observed for the period 1912-2017.

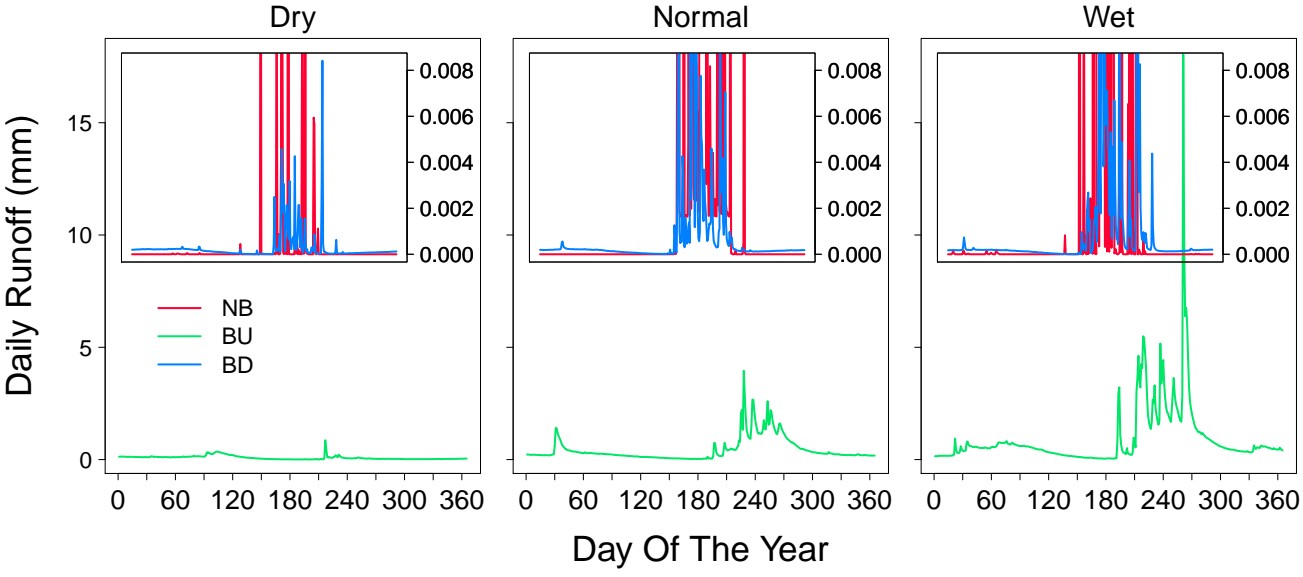

**Figure 3.** Observed annual discharge characteristics for the three different flow categories as defined in Section 2.2 for a typical dry, normal and wet year.



**Figure 4.** Upper panel shows the observed mean daily discharge for station 11 in Fig. 1) for 2000–2011. Middle and lower panel show the identified discharge percentile and the total number of days per year within drought, respectively, as estimated from the combined method as proposed by Van Huijgevoort et al. (2012) (Old Method) and the using modified combined method proposed here (New method)





**Figure 5.** Comparison of percentile and duration between old method and new method for daily discharge observation from station 11 in Fig. 1 for 2000–2011





**Figure 6.** For three different stations belonging to each runoff category, the hydrograph (black) and corresponding discharge percentile are presented. Also shown at the bottom of each panel is the occurrence and intensity of a drought or wet spell.



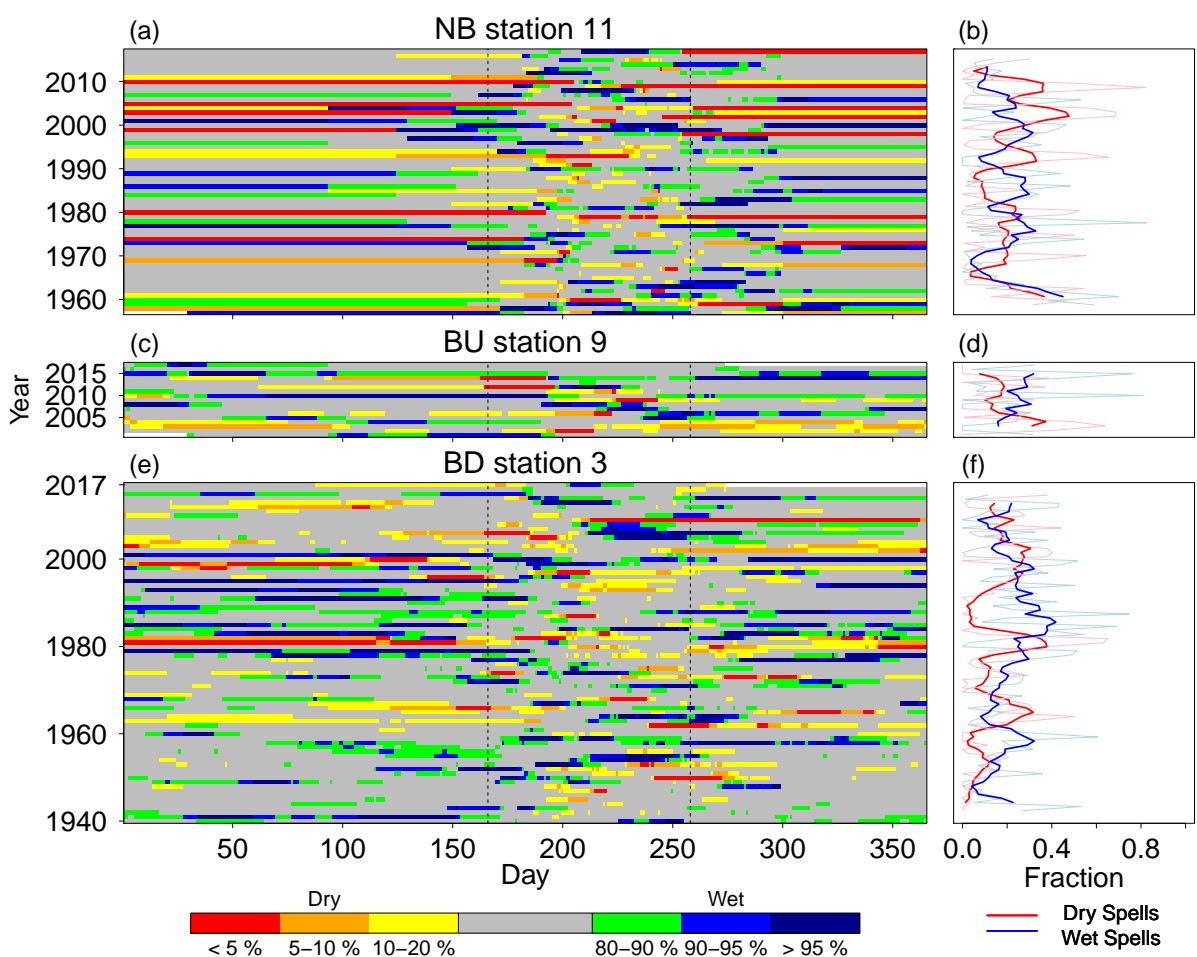

**Figure 7.** Left column shows the intensity and occurrence of hydrological drought and wet spell for three different stations belonging to each runoff category. Dashed vertical lines indicate the start and end day of the NAM. The right column shows for each year the number of drought and wet spell days (both original data and 5 year moving average).





**Figure 8.** For the identified drought, panel a) presents the starting date and duration of a drought for each basin corresponding to a given runoff category (see Section 2.2). Panel b) shows the distribution of the start date of a drought for a given season, while panel c) a boxplot of the drought duration of a drought starting in a given season for a given runoff category. Panel d)–f) present similar results for identified wet spells.





**Figure 9.** Influence of different values for the upper (Pu) and lower (Pl) percentile threshold and varying the inter-event time ($N$). Legend in bottom-right corner indicate the total number of observed drought or wetspells for each inter-event time.



**Figure 10.** The impact of using different moving average (MA) windows from 30 days to 5 years on the start data and duration of a drought or wet spell.



**Figure 11.** Similar to Fig. 10, for a MA period of one year for the three different runoff categories used here.





**Figure 12.** For each station within each runoff category the mean drought (a) and mean wet spell (b) duration as a function of NAM discharge fraction is shown.





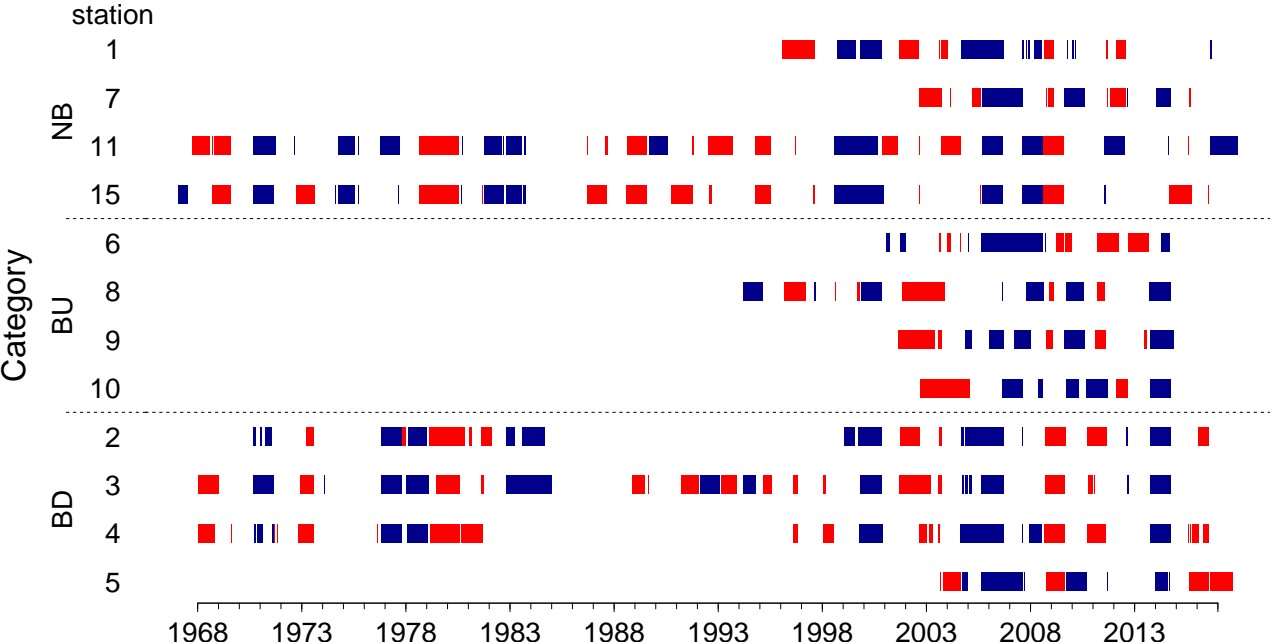

**Figure 13.** Observed hydrological drought and wet spells for the 12 different locations during the last 50 years using a one-year MA window.