# Peer review of "The impact of elevation and flow dynamics on hydrological drought and wet spell characteristics in semi-arid southeast Arizona"

_Hydrology and Earth System Sciences, 2019_

## Referee Comment (RC1) · Anonymous Referee #1 · 30 Sep 2019

**Review of "The impact of elevation and flow dynamics on hydrological drought and wet spell characteristics in semi-arid southeast Arizona" by Lu et al.**

The authors modify the combined threshold level method and consecutive dry period method, originally proposed by Van Huijgevoort (2012), to be more applicable for regions with strong precipitation seasonality (e.g., monsoon season). In the modified version of the method, consecutive dry / drought day values were compared with the same variable for all other years for each calendar day seperately (and not with events as was the case for the old method). By doing so, the drought identification method allows droughts to persist from the wet to the dry season. Additionally, the drought identification method results in a uniform percentile distribution for each calendar day. The authors applied the method to (tributaries of) a case study river in Arizona (US), and reveal more satisfying identification of streamflow drought events, as well as clear differences in drought occurrence and characteristic between the main river branch (without base flow) and upstream tributaries for which base flow plays a more important role.

In general, I like the modification of the methods as it allows for a more consistent comparison of drought over space and time, e.g., the 20$^{th}$ percentile threshold is exceeded 20% of time for each DOY and (sub-)catchments. On the other hand, there is one aspect of the methods that I did not find very satisfying, and I am curious to the opinion of the authors. In addition, there are some other major and quite some minor comments that should be addressed before the manuscript is suitable for publication.

Major comments.

1. The proposed method and modifications are definitely an improvement. However, there might be one more point of consideration. With the suggested method, you kind of force droughts to start in the flow season. Droughts starting in the dry season often have to "catch-up" with droughts starting in the flow season, as their consecutive number of drought/dry days is lacking behind. However, even when their consecutive dry day number is lacking behind, their total period of zero flow might last way longer then the drought that started in the wet/flow season. The official start of the lacking-behind dry season drought can only occur when some earlier starting droughts (in other years) are terminated. In other words, the official start point of a drought starting in the dry season might be after a prolonged period of zero discharge. I believe that a more optimal definition would take into account the total period of zero flow, and:
   o Encourage the authors to come up with another modification of the method that somehow take into account the total length of the dry spell or provide an argument why the do not think this is needed.
   o *An option might be to assign each zero flow day the value of the total length of the dry period. So, if you have a dry period of 10 days, you give each day a value of 10, rather than a value from 1:10. You can do a similar percentile ranking based on these values. This might work....*
   o In any case, the authors should definitely investigate how long a period of zero flow can occur before a "drought" starts and discuss this.
2. I engorge the authors to be more careful with the use of the wording "hydrological extremes" throughout the manuscript. They mostly look at **anomalies**, not extremes. This is an important distinction to make upfront (especially with regard to the communication of the results to a broader audience), especially because some of the "extremes" are not "as extreme" (see next comment)
3. Figure 6 (upper panel) reveals that only a very small discharge event (only visible when completely zoomed-in) triggers the identification of a wet period. This period then last for a

30 days as the authors apply a 30-day moving average. I agree that this one-day <0.1 mm discharge event can be abnormal, but to classify this entire period as wet might not be the most suitable.

    a. The authors should discuss this, especially in the context of the term "hydrological extremes" used throughout the manuscript.

    b. In addition, droughts might develop more gradually then wet periods. Could the authors comment on whether the use of a 30-day moving average window is equally suitable to both drought and wet period?

4. I do not agree that one should definitely correct for human induced step changes and trends for drought analyses (Section 3.3). From an instream perspective, human induced changes and modifications in drought characteristics can be of critical importance as well. In addition, the proposed method of recalculating drought characteristics for different reference periods (before and after change points, where the change point might vary depending on the river), might hinder a fair comparison of drought characteristics over space and time, as each time a different reference period is used. So, in case the authors want to correct for trends and step changes, it should be better justified why (in Section 3.3), and the consequences should be discussed.

5. I like the consistency of the proposed method, i.e., the 20$^{th}$ percentile is exceeded 20% of the time. However, later on, the authors apply different pooling approaches. Did the authors test how these approaches modify the amount of time anomalies occur for each catchment and DOY?

Minor comments:

The manuscript requires some more thorough editing. Suggestions (and some other points) in the minor comments below.

- **Title:** a major part of the paper deals with modifying the combined TLM and CDPM. I think this should somehow be reflected in the title.
- Line 3: regularly occurring no flow conditions instead of "no flow conditions"
- Line 5: The wording "hydrological extremes". I would be very careful with this wording throughout the manuscript. You mainly identify anomalies, not extremes.
- Line 16: the abstract does not mention anything specific about the impact of elevation although it is in the title. Maybe it can be added here.
- Line 17: correct the spelling of "drought and wet spells patterns"
- Line 18: wet spell**s** and drought**s** instead of "wet spell and drought are rare".
- Introduction: I strongly encourage the authors to review some more recent literature, especially about the application of the combined method between 2012 and now.
- Introduction: be more careful in your referencing, e.g.: Line 25: The sentence about the costs of drought with a reference to Dai (2011) is also in the Van Huijgenvoort paper. However, they refer to a different contribution of Dai (2011).
- Line 38: "severe dry spells impacting the land surface". You could be clearer here.
- Line 38: Where does "This" refer to?
- Line 41: if you focus on both dry and wet spells, it would be good to give them equal amount of attention. Now most focus on droughts.
- Line 43: remove "(" after e.g.
- Line 50: You could add to this argument that some studies kick-out stations with regularly, occurring zero flows. (I am guilty as well; Tijdeman et al., 2016).
- Line 51: "incorrect higher threshold". Why is a higher threshold incorrect? What is a correct threshold?

- Line 54: if true, the authors could add that the CDPM was developed for precipitation.
- Line 56: The main problem of the method was the difficulty to characterize the transition from flow to low-flow periods, or? Not the identification of the start.
- Line 61: elevation is not a direct control. In this study, elevation encompasses a variety of different controls such related to snow, the occurrence of groundwater, and the occurrence of local rainfall events.
- Line 68: "To limit the impact of human influences" I would remove this from the sentence as it reads as one of the goals from the study.
- Line 82: suggest to remove "detailed"
- Line 83: suggest to remove "detailed"
- Line 86: "as explained in the introduction" If you explain something in the introduction, you do not have to repeat.
- Line 92: "300-350 mm of precipitation" → per year?
- Line 110: Be consistent. You use both stream flow and streamflow.
- Line 118: spell instead of "spells"
- Line 134: you could also refer to the threshold level method here.
- Line 135: you could explain why these methods do not work well in these domains.
- Line 138: More careful phrasing: I doubt that this is the most widely used method, especially not for wet "hydrological extremes".
- Line 139-141: More careful phrasing needed as this is not nescecairly true. Some (of the mentioned) studies methods use daily flow values to define a monthly or annual threshold (e.g., fixed threshold).
- Line 143: All referenced studies are drought studies. Any example of the use of the TLM to identify wet spells?
- Line 143-144: the threshold level method can never distinguish between a dry period and drought. The problem with regularly occurring zero flows:
    o Droughts are indicated for all zero flows or for none of the zero flows, depending how you define your quantiles.
    o It is quite complicated to rank zero flow events, i.e., how do you rank five zeros (see Stagge et al. (2015) how they deal with zero precipitation in the SPI).
    o These problems should be presented more clearly (either here or in the introduction), as you are going to solve them later.
- Line 162: I would describe the rescaling method more clearly.
- Line 164: ">20%" this is not a wet spell.
- Line 166: "Major advancement" → Any references to studies that applied this method would be nice her. Here, or in the introduction.
- Line 204: why 25 and 75? Why not 20 and 80 (your drought and wet spell thresholds).
- Line 182-187: Maybe use bullet points here as well. This would highlight the modifications you made.
- In general, I would strongly encourage the authors to use some variables and equations in both the new and modified method description (as is done in the prev. paper, where it increased the clarity).
- Section 3.3. A reference to Sadri et al., 2016 (HESS) would fit nice here (uses same kind of tests to identify human modifications of low flow).
- Line 200: "as indicated in the introduction …" -> repetitive and not needed.
- Line 202: "is necessary to identify and correct for these impacts" do not agree (see major comment (3). Please elaborate why.
- Line 224: "The previous section presented …" → suggest to delete.

- One missing first result section would be the evaluation of the newly proposed method / comparison between the new and old method. (4.1). Now, this is more or less done in the methods section. However, I believe this better fits the results.
- Line 231: The fact that you apply a moving average on your data should be presented earlier (before the introduction of the combined method).
- Line 233-234: Did the authors recalculate the threshold after applying a different moving average interval?
- Line 235: Introduce abbreviation "PDSI"
- Define duration in the methods and be consistent. Is this the total time in drought for a certain year (as x-axis Figure 4c) or the length of the drought / dry spell (e.g. Figure 8).
- There are many unnecessary repetitions at the start of each new section (e.g., the first five lines of 4.3). I would suggest deleting these to improve readability.
- How are probabilities in Figure 8, 9, 10 etc. calculated. And can the authors confirm that the dips in probabilities at the left and right side of Figure 8b,e are actually there? And not an artefact of any kind of smoothing that does not take into account observation before and after the plotted period?
- Line 251-255: This is repetition and could be deleted.
- Line 257: As indicated in Section 2.2 … (repetition).
- Line 258: spells instead of "spell"
- Line 258: I could not find a reference to Figure 5 before (and after) you introduce Figure 6.
- Line 268: Discharge instead of "Moisture"
- Line 285: For both stations?
- Line 301: For the representative station or for all stations?
- Line 304: I would not call a 100 days "very short"
- Line 346-350: This is all repetition, which the authors could consider to remove.
- Line 355-356: to much brackets, especially to much ")" …
- Line 368: Use of wording "hydrological extremes" (see earlier comment)
- Results general: there is some discussion in the results (e.g., 295-300). I would suggest the authors to more strictly separate.
- Discussion general: too much repletion of the results and too less interpretation. For example, various other studies show the effect of pooling and smoothing, and the authors could relate their findings to these works (+ a recommendation which smoothing method to use). In addition, there are also quite some new results in the discussion. I suggest separating.
- Line 408: "impact of elevation and flow conditions." Elevation does not have a direct impact (and "flow conditions" is a bit of a vague term).
- Line 426: "dominant driver" of what?
- Line 432: "short duration hydrological extremes" … again not extreme. Anomaly.
- Figure 2: You might add the trend from 1960-onward to the legend to show that it is less or absent.
- Figure 2: Location nr. 2 has only data from 1967 –onward (according to table 1). Location 3 or error in table 1?
- Figure 3: I have difficulties to compare the timeseries in the panel plots. It might be better to separate them? And maybe use a log-scale, so you can see higher flows as well?
- Figure 2,3,4,5: be consistent. You now use discharge (mm) vs. runoff (mm) vs. discharge (m3/sec) vs. moving average discharged(?) in Figure 5.
- Figure 4c, x-axis: duration or days in drought?
- Figure 4, caption: remove ")"
- Figure 7:  I would not use green for wet spells (subjective opinion). Maybe light blue as is done in the USGS WaterWatch? https://waterwatch.usgs.gov/

- Figure 8: I would separate the probability plots from the main plots, as these are not really related.
- Figure 8: do the lower probabilities on the right and left side of the probability plot indicate a lower probability of occurrence (e.g., left of Fig. e)? Or is this some kind of a smoothing effect (due to a possible absence of points at the start and the end of the year).
- Figure 8: modify the plotting area to fit all observations.
- Figure 8-11: What is the difference between the probability (8-9) and density (10-11)?
- Figure 11: Why a MA period of 1 year? If you apply a one-year MA, you probably remove all zero values and are comparing results for the TLM.
- Figure 12: Any pooling applied here?
- Figure 13: For similar reasons: why a 1-year MA average window.

Stagge, James H., et al. "Candidate distributions for climatological drought indices (SPI and SPEI)." International Journal of Climatology 35.13 (2015): 4027-4040.

Sadri, S., Kam, J., and Sheffield, J.: Nonstationarity of low flows and their timing in the eastern United States, Hydrol. Earth Syst. Sci., 20, 633–649, https://doi.org/10.5194/hess-20-633-2016, 2016.

Tijdeman, E., Bachmair, S., and Stahl, K.: Controls on hydrologic drought duration in near-natural streamflow in Europe and the USA, Hydrol. Earth Syst. Sci., 20, 4043–4059, https://doi.org/10.5194/hess-20-4043-2016, 2016.

---

## Referee Comment (RC2) · Anonymous Referee #2 · 10 Oct 2019

The authors deal with a modification on the definition of hydrological drought that is supposed to be more robust for ephemeral rivers with prolonged zero-flow periods. While I understand the reasons behind the need of such modification of the van Hui-jigevoort et al. method, I find the message not well communicated and the proposed solution not fully on point. Both the tackled issue and the proposed solution need, in my opinion, to be better described, maybe with the help of a visual representation of a real study case (for a specific event/year) or even an artificial case (that highlight the key drawback of the current method) before showing the performance on all the available data. At the current state, it is difficult to gasp how the suggested modification actually works, since the full description is based only on text, and the examples in Fig. 4 is

really hard to read. Also, since the method is needed for zero-flow rivers, the authors should focus only on such stations rather than all the available stations. If I understand correctly, the authors state that the method of van Huijigevoort et al. may "break" a drought event in two events if an event start during the wet period and continue during the dry one, but this should be highlighted in a figure that show one of such case and how the method solve the issue. My understanding is that the author define a cdf of number of (antecedent) dry days that is different for each day of the year, rather than the same for all the day based on the total length of the zero-flow. This seems a solution to avoid "breaks" for events that started during the wet period, but can lead to difficulties for events starting at the beginning of a dry period. If my understanding is correct, I suggest to the authors to consider, first of all, if their goal is to produce an indicator that can be update in near real time (while the event is developing) or that defines the drought on past data. In the second case, better solutions can be found than the one proposed. Since a "true" definition of drought/wet spell start and length is not available (for obvious reasons), the authors need to clearly highlight that their outcomes are at least more reasonable that the one obtained with the previous method and not just as arbitrary. A second point of contention for me is the need for a better explanation on the reasoning behind this kind of definition of hydrological drought and (especially) wet spells in zero-flow rivers. As the authors stated, they are looking at an issue that arise for specific rivers, with zero-flow during most of the year and flow only during monsoon, but it is not clear what is the goal of having hydrological drought (or wet spells) defined for such rivers in such a way. While the classic analysis of dry spells (length of periods of zero-flow) or wet spells (length of period with positive flow) in such rivers is relevant, I do not understand, for instance, the reasoning to define a day as part of a "wet spell" even if the flow is zero (e.g., first half of 2007 in Fig. 6a). Finally, the title of the paper is ambiguous, since the focus in on a redefinition of drought events in zero-flow rivers but the title seems to imply that the effects of elevation and flow dynamics in semi-arid rivers will be discussed. Follow some minor comments: P2L30. Why wet spells on rivers are defined based on precipitation here? P2L39. This is not

true for all the cases. There are plenty of evidences that over some regions less extreme drought are expected (e.g., northern Europe). P2L50. What is the relevance for water managers in rivers with zero-flow? Are those rivers under any water managing? P4L88. Why the river is here identified as perennial, whereas is defined ephemeral in the rest of the text (see e.g., P3L74) or just partially perennial (P4L103). Please be consistent. P4L120. You should focus only on the NB rivers, and eventually show that your method works for the other rivers too (if this is the case). P6L142. This should read as: "The TLM has a problem for locations with zero flow (in a specific period) for a considerable amount of years". Please reword. P7L181. It is really difficult to extrapolate how the two methods work from fig. 4b. If 2003-2004 is a good example year, please make a specific figure that highlight the key differences between the two methods. P8. Sensitivity Analysis. This section does not seem well thought-out in my opinion. The range of values adopted in this analysis need to be better supported by some reasoning (e.g., pooling up to 180 days? moving windows of 5 years?). P9L265. This description of the behavior of drought is a consequence of your definition of the events rather a fact. You need "independent" evidences on the behavior of drought to support that your reconstruction is more adherent to the reality than the one obtained with the previous method. P10L290-300. This analysis on the long-term variations is out of topic and not well supported by formal trend tests and analyses. P11. Fig. 8. This figure is rather confusion, and, in my opinion, not the best way to convey the key finding that the authors want to show here. P12. Section 4.3 is this on only one river station or all the stations combined? This is not clear. P13L374. If a backward moving window is used (rather than a most common centered one) this need to be clarified and justified in the methodology.

---

## Referee Comment (RC3) · Anonymous Referee #3 · 4 Nov 2019

This manuscript deals with a new methodology for assessing hydrological drought with a variable threshold approach in semi-arid catchments and its application on case study catchments in Arizona.

**Main comments**

The comments will clearly overlap with those of the other referees, emphasizing the main issues found in the manuscript. But before that, it has to be noted that the study is reasonably well conducted and the manuscript generally well written and organised.

However, several major issues can be identified:

1. The study extends some previous work on a generic method for defining a hydro-logical drought with a variable threshold (Van Huijgevoort et al., 2012), a method that would be applicable to semi-arid catchments with long periods of zero flows. First, the methodological extension presented L165-L198 is not clearly described, and would deserve some more didactic illustration and/or pseudo-algorithm.

2. Second, one wonders about the usefulness of proposing an extension of the variable-threshold hydrological drought definition. Indeed, the basic reasoning of using a variable threshold is to detect streamflow lower than usual. When streamflow is usually zero, what is lower than usual? The conceptual limits of the variable threshold method are here clearly reached. Indeed, what is the point of by all means trying to compute/define a streamflow deficit when there is no streamflow? I guess that defining drought in a non-perennial river may be useful, for example for aquatic biodiversity, but streamflow drought is here not relevant. What could be relevant for example in this case is defining an edaphic drought in the river bed to assess the decline of soil moisture during the dry season, and the state of the corresponding habitat for invertebrates.

3. Third, and in relation to the above comment, the objective of the study is clearly ambiguous. Indeed, the title does not even mention the proposition of a new methodology. Second, the current title suggests some relationships between catchment characteristics and drought characteristics, and the way results are presented – including the poor map of Figure 1 that don't even show the delin-eation of catchments – do not allow to even extrapolate results in other basins with similar characteristics.

4. Fourth, the interpretation of results in terms of hydrological droughts as computed with the newly proposed method is in my view just that: an interpretation. Indeed,

this is when results are presented that the reader realises that there is no possible assessment of whether the new method is more relevant than another one, because precisely of the questionable relevance of using a variable threshold for zero flows. This method indeed comes down to assess how much the river is not flowing...

As a conclusion, a new methodology to compute hydrological drought – understood as anomalies with respect to a variable threshold – for semi-arid basins has to show a continuity with perennial basins in order to be taken as a serious candidate. This would in my opinion be the only way to assess the underlying computation assumptions. This is what the authors have tried to present, but results are unfortunately unconvincing. And even with such a continuity, I definitely question the overall approach of hydrological drought understood as anomalies when considering non-perennial basins. This study has indeed one merit: to exemplify the irrelevance of considering variable threshold approaches for characterizing hydrological droughts. As an example on the other side of the hydrological spectrum, I am not sure why a slightly-shorter-than-usual several-month-long period with a dry riverbed would be named a wet spell for a given basin.

**References**

Van Huijgevoort, M. H. J., Hazenberg, P., Van Lanen, H. A. J., and Uijlenhoet, R.: A generic method for hydrological drought identification across different climate regions, Hydrol. Earth Syst. Sci., 16, 2437–2451, https://doi.org/10.5194/hess-16-2437-2012, 2012.

---

## Author Comment (AC1) · 11 Feb 2020

***bold indicates our response.

Reviewer 1

Our general response

The original manuscript submission did a poor job delivering a number of messages and providing additional details. We have tried to address these in an updated version of the manuscript. The main comments raised by all three reviewers can be summarized in the following three major concerns:

1. The manuscript lacks a strong motivation why one would be interested to focus on drought and wet spells within a semi-arid environment that is predominantly dry throughout the year.
2. A good description of the newly presented combined identification method including a logical figure.
3. An incorrect focus where ephemeral rivers were compared with perennial rivers, without providing the necessary relationships.

We have tried to address these major issues in a new version of the manuscript and we believe that this has strongly improved the manuscript readability and clarifies the above mentioned topics. This original submission was poorly presented and we would like to thank the reviewers for identifying this. However, we also believe that the updated version provides considerable interesting insights into the occurrence and variability of semi-arid hydrological anomalies that strongly reflect the ecohydrological functioning of the channel bed and riparian zone, which we believe are of interest to readers of HESS.

Major concern 1:

**To the introduction we have added the following:**

**"This combined procedure was able to identify hydrological anomalies specifically within transitional regions, where zero flow conditions are common but not the standard. However, as the current work will show, this method has difficulty correctly identifying drought and wet spell occurrences within ephemeral rivers within semi-arid regions with a strong seasonal precipitation signal (e.g. Monsoon). The strength and occurrence of seasonal runoff within these rivers strongly impacts groundwater recharge, and the ecohydrological state of the channel bed and riparian zone (Goodrich et al., 2004; Scott et al., 2008). As a result, these reaches become hotspots for biodiversity, especially during the dry season (e.g. Moreno-de las Heras et al., 2012; Cleverly et al., 2016). Correct identification of a hydrological anomaly is therefore important. As a result, this paper presents an updated version of the combined approach originally developed by Van Huijgevoort et al. (2012), improving the identification and continuation of a drought or wet spell, when transitioning from the wet into the dry season. This newly combined approach provides information on the hydrological status of the river including the occurrence of anamolies. For semi-arid ephemeral channels this information serves as a proxy for the moisture state of the channel bed and riparian zone, and its ecohydrological functioning (Scott et al., 2008; Moreno de las Heras et al., 2012). Specifically, this paper focuses on the semi-arid southwest US, where considerable amounts of long-term observations are available, which allows for a detailed analysis."**

**To Section 2.1 we have added the following:**

**"Within the San Pedro, upslope headwaters have low permeability bedrock relatively close to the surface, resulting in infiltrating rainfall and snow melt quickly to reach the groundwater system and river network (Fan et al., 2007; Kampf et al., 2016). These perennial streams become ephemeral downslope through transmission losses from evaporation and infiltration into the dryer ambient subsurface (Cataldo et al., 2010; Blasch et al., 2013). Within the semi-arid San Pedro basin, stream infiltration has been shown to account for 10-40% of total groundwater recharge (Goodrich et al., 2004). However, for the majority of precipitation generating flow events the depth of infiltration is relatively shallow, with only the biggest events infiltrating deeper (>1 meter below the channel bed). Transmission losses therefore strongly impact root zone moisture availability. Lower elevation riverbeds and their**

surrounding areas are therefore favorable for biodiversity (e.g. Moreno-de las Heras et al., 2012; Cleverly et al., 2016)."

**To Section 3.2 we have added the following:**

"As indicated in the introduction, flow events replenish rootzone moisture below the channel bed, which is used by the riparian zone vegetation during the dry season for transpiration. Therefore, the occurrence of a drought or wet spell during the positive flow season, directly impacts moisture availability afterwards. As such, it is important that a given hydrological state continuous from the wet into the dry season. For a hydrological drought to prolong from the wet into the dry season, drought should also be identified at lower consecutive drought/dry day numbers, which currently is impossible."

**To the discussion we added the following:**

"It should be noted that for situations where there is generally no flow, a small flow event can result in a sudden increase of the discharge percentile, as shown in Fig. 6 for the early summer of 2007. This can result in a wet spell anomaly, which due to a 30-day MA window size, can last for 30 days even though the total flow amount during this period was very small. As the channel flow ceases, the discharge percentile becomes more indicative for the moisture state of the channel bed, which can be expected to be relatively wet due to the strong impact of transmission losses."

Where we felt needed, we refer to these sections throughout the rest of the manuscript. By providing these details we believe the motivation and focus behind this work become clear to the reader.

Major concern 2:

To address this point, in Section 3.2 we have tried to identify the current limitation of the original combined method of Van Huijgevoort et al. (2012). Furthermore, in our original submission we did not highlight that this method was only developed for the identification of drought. Also, the updated version provides now a link to point 1 raised above, to highlight that a given hydrological state as well as the occurrent of a drought and wet spell, here reflects that available root zone moisture.

Therefore, Section 3.2 was completely rewritten. The first part now contains the following:

"Even though the combined method proposed by Van Huijgevoort et al. (2012) has been a major advancement to identify hydrological anomalies within transition regions (e.g. Breyer et al., 2018; Heinke et al., 2019), the method has three potential issues. First, the final discharge percentile distribution for a given day of the year does not have to be perfectly uniform between 0–100. This is the result of combining the original TLM, where discharge percentiles are obtained for a given day, with the CDPM, which defines the cumulative density function (cdf) on the bases of all dry days. Even though consecutive drought/dry numbers are rescaled (see step 3 of Section 3.1), this does not guarantee a perfect uniform distribution between 0 and 100 for a given day.
Second, during zero flow situations, drought only occurs in case a given day has a high consecutive drought/dry day number. For the San Pedro these occur in spring at the end of the dry season. Therefore, in case a drought starts during the positive flow season in summer, it will generally stop identifying the drought once flow ceases. If zero flow conditions continue to occur, the combined method will then identify a drought again later in the dry season in spring. As such, the original combined method therefore identifies two droughts (i.e. one in summer during the NAM and one in spring after a prolonged period without flow, see also Section 4.1). As indicated in the introduction, flow events replenish rootzone moisture below the channel bed, which is used by the riparian zone vegetation during the dry season for transpiration. Therefore, the occurrence of a drought or wet spell during the positive flow season, directly impacts moisture availability afterwards. As such, it is important that a given

hydrological state continuous from the wet into the dry season. In order for a hydrological drought to prolong from the wet into the dry season, drought should also be identified at lower consecutive drought/dry day numbers, which currently is impossible.

Third, originally the method was not developed to identify wet spells for zero flow conditions, as these would always occur during situations of low consecutive drought/dry day number. For the San Pedro, this would result in a wet spell to occur immediately at the start of the dry season, even if a drought was observed during the positive flow season (see also Section 4.1)."

We also extended the explanation of the new combined procedure presented, following an approach similar to the one presented in the original work of Van Huijgevoort et al (2012) making use of both mathematical symbols and enumerated lists. Here we use a figure to indicate the different steps.

Furthermore, in line with suggestion raised by the reviewer, we moved the figure that showed the difference between the old and new combined method to a newly created Section 4.1, which contains the following:

"4.1 Comparison of the original and modified combined method

Figure 5 shows the difference in identified discharge percentile between the combined method as proposed by Van Huijgevoort et al. (2012) (old method) and the modified combined method proposed here (new method). As indicated in Section 3.2 the old method was unable to identify drought during zero flow conditions with a low consecutive drought/dry day number. This can clearly be observed from Fig. 5b where these days, that generally occur at the beginning of the dry season during the fall, show high corresponding discharge percentiles. Furthermore, for the years 2004–2005 and 2009–2010, the drought observed during the positive flow season, first ceases and return the following spring at the end of the dry season for high consecutive drought/dry day number. This situation does not occur for the new method presented here, where the consecutive drought/dry day number of a given zero flow day is compared with its number observed during other years. As such, discharge percentiles in the fall can be small enough to indicate a drought. As a result, for both 2004–2005 and 2009–2010 the drought continued from the wet season into the fall, as an indication of the moisture state of the riparian zone. The fact that the observed hydrological drought continues after precipitation ceases results in a considerable increase in the total number of drought days for 2002–2005 and 2009–2011 (Fig. 5c).

Section 3.2 also mentioned that, the original method of Van Huijgevoort et al. (2012) would lead to high discharge percentiles at low consecutive drought/dry day number and as such was not specifically developed for this. For the fall, Fig. 5b shows for station 11 this would indicate the occurrence of a wet spell during the fall of almost each year, irrespective of the strength of the NAM. The newly combined method presented here, does not show this strong dependency. In stead, high discharge percentiles in the fall only occur during years where already during the NAM a wet spell was observed (i.e. 2000–2001 and 2006–2007). During the other years discharge percentiles in the fall do not indicate the occurrence of a wet spell."

We believe these changes have improved the description of the new combined procedure presented here and its applicability to identify hydrological anomalies within semi-arid ephemeral channels.

Major concerning 3:

As the current work focuses on hydrological anomalies within semi-arid environments and more specifically the San Pedro basin, we feel that only focusing on the ephemeral channels as suggested by reviewer 2 would not provide a complete picture of hydrological anomalies occurring throughout this region. Instead we assess their occurrence of three typical cases: 1) dry lands channel beds that only show runoff from intense precipitation events during the NAM and from remnants from hurricanes in the fall (NB), 2) the upland regions that have perennially flow conditions (BU), and 3) low land regions with perennially flowing conditions received from baseflow upslope (BD). To introduce this we added the following to the introduction:

"Results obtained for ephemeral channels as derived using the new combined procedure presented here, will be compared with observations from perennial rivers at higher elevation

as well as for downslope locations receiving continuous flow from upslope. By analyzing the occurrence of drought and wet spell anomalies across these locations, this work will provide a detailed overview of their occurrence and variability within the San Pedro, as well as the role of climate and local geographical location."

For case a) dry lands channels, the occurrence of a single event can generate a wet spell even after runoff ceases. This might be counterintuitive, as indicated by all three reviewers, as "How can zero flow conditions correspond to a wet spell". We hope that the comments provided by major point 1 and 2 as given above, have resolved this aspect.

Reviewer 2 indicated that one option would be to only focus on case a). As we did a poor job describing the different system, we understand this suggestion. However, we hope by correctly adding these details throughout the paper, it becomes clear why it is important also to address the uplands.

For case b) the hydrological response of these the upslope regions is much more similar to temperate environment. Hydrological drought and wet spells have a direct link to the amount of water available in the river network. Furthermore, as flow conditions are always positive, we added the following to Section 3.2:

"for locations within continuously flowing condition (BU and BD category) the TLM approach was used solely."

For case c), at shorter time scales (30-day moving average window), observed hydrological drought and wet spell characteristics resemble those observed for the upland reaches, since during the dry season all water originates from these uplands. However, also in these environments, transmission losses form an important source of moisture for the surrounding riparian zone. Since these channels transport the majority of their water during the NAM, when focusing on a longer timescale (e.g. MA-window of 1 year), their hydrological drought and wet spell characteristics resemble more case a) the NB category. As such, at these timescales a hydrological anomaly is more representative for the state of the riparian zone. Again, the paper did a poor job in explaining this.

Because of this difference in behavior across MA-window scales, we felt it was important to assess this. Therefore, we presented two figures for 30-day MA windows (Figs 6-8), various MA-window sizes (Fig. 10) and a one-year MA-window size (Figs 9 and 11). However, we did not properly motivate these choices and have tried to include this in the updated version of the paper, by addressing these in both the Results section as well in within the Discussion. The latter now states:
"The mean duration of a wet spell last longer for upslope domains (BU) as compared to the lower elevation categories (BD and NB). For the upslope regions, shallow groundwater is expected to have a stronger control on the observed amount of baseflow for a longer period of time, increasing the mean duration (see also Section 4.2)."

Besides addressing these major concerns we have:

- Changed the title as suggested by the reviewers
- Include more up to date references in introduction and section 3
- We propose to alter the setup of the manuscript by creating a new paragraph 4.1, before the original sections 4.1-4-3 which will become section 4.2-4.4.
- For the one-year MA-window analyses in Figures 11 and 13, we have added the details of why we feel this is interesting both in the Introduction as well as in Section 3.4 and in the Discussion. We will also highlight that this situation effectively corresponds to applying the TLM only.
- To decrease the length of the discussion and not to present new results we have moved the figures of the discussion of the original manuscript to a newly created section 4.5.
- Adjust the original figures where suggested.

**We are convinced that by resolving these concerns and addressing the various issues raised by the reviewers the quality of the manuscript has improved consirably.**

**Review of "The impact of elevation and flow dynamics on hydrological drought and wet spell characteristics in semi-arid southeast Arizona" by Lu et al.**

The authors modify the combined threshold level method and consecutive dry period method, originally proposed by Van Huijgevoort (2012), to be more applicable for regions with strong precipitation seasonality (e.g., monsoon season). In the modified version of the method, consecutive dry / drought day values were compared with the same variable for all other years for each calendar day seperately (and not with events as was the case for the old method). By doing so, the drought identification method allows droughts to persist from the wet to the dry season. Additionally, the drought identification method results in a uniform percentile distribution for each calendar day. The authors applied the method to (tributaries of) a case study river in Arizona (US), and reveal more satisfying identification of streamflow drought events, as well as clear differences in drought occurrence and characteristic between the main river branch (without base flow) and upstream tributaries for which base flow plays a more important role.

In general, I like the modification of the methods as it allows for a more consistent comparison of drought over space and time, e.g., the 20th percentile threshold is exceeded 20% of time for each DOY and (sub-)catchments. On the other hand, there is one aspect of the methods that I did not find very satisfying, and I am curious to the opinion of the authors. In addition, there are some other major and quite some minor comments that should be addressed before the manuscript is suitable for publication.

**We would like to thank reviewer 1 for the constructive comments. This really helped us to evaluate the quality of our manuscript an assess the current bottle necks.** Major comments.

1. The proposed method and modifications are definitely an improvement. However, there might be one more point of consideration. With the suggested method, you kind of force droughts to start in the flow season. Droughts starting in the dry season often have to "catch-up" with droughts starting in the flow season, as their consecutive number of drought/dry days is lacking behind. However, even when their consecutive dry day number is lacking behind, their total period of zero flow might last way longer then the drought that started in the wet/flow season. The official start of the lacking-behind dry season drought can only occur when some earlier starting droughts (in other years) are terminated. In other words, the official start point of a drought starting in the dry season might be after a prolonged period of zero discharge. I believe that a more optimal definition would take into account the total period of zero flow, and:
o Encourage the authors to come up with another modification of the method that somehow take into account the total length of the dry spell or provide an argument why the do not think this is needed.
o *An option might be to assign each zero flow day the value of the total length of the dry period. So, if you have a dry period of 10 days, you give each day a value of 10, rather than a value from 1:10. You can do a similar percentile ranking based on these values. This might work….*
o In any case, the authors should definitely investigate how long a period of zero flow can occur before a "drought" starts and discuss this.

**Thanks for the suggestion. We agree with the reviewer that for the old approach as proposed by Van Huijgevoort et al. (2012) it is indeed the case that the dry season often has to "catch-up" with a given drought due to the low consecutive number drought/dry day number. The improved method as proposed in the current work was specifically developed to improve this situation. We have updated figure 5 of the manuscript which now shows the year 1973. This figure shows that for new approach, this "catching up" is not occurring anymore. We agree with the reviewer that the original manuscript did not emphasize this aspect. Therefore, we have decided to stress this aspect more by adding additional text in Section 3.2 of the manuscript.**

[Figure]

We very much like the suggestion of the reviewer to consider the total length of the dry spell. This took a little while to properly implement, but the result, in comparison with the method presented in the paper are shown in the figure below. These new results were added to Figure 5 of the original manuscript as shown below, which in the new submission will be treated in a new Section 4.1. The upper panel shows the observed mean daily discharge for Flume 1 of WGEW (station 11 in Fig. 1) for 2000-2011, while the middle and lower panel show the identified discharge percentile and the total number of days per year within drought, respectively. The lower panels show the combined method as proposed by van Huijgevoort et al. (2012) (Old Method), the using of modified combined method (New method) and the reviewer recommended method(Reviewer's method).

In panel b) we can observe that while the percentiles of the old and the new method gradually decrease, using the total length of the dry period leads to a drastic decrease in the discharge percentile at the start of the dry season that subsequently stays low at a constant value for a relatively long period of time. As such, this total dry period length based method is not continuously changing. In panel c), we could see that compared to the new method, in year 2001 and 2002, reviewer's method identified longer duration of droughts while in year 2003 and 2004 it identified shorter duration of droughts.

A main reason why we proposed the new approach presented in this paper is that it allows a given drought that occurs during the monsoon season to continue as runoff ceases. As this drought is a proxy for the available moisture within the channel bed. For the total length based approach, this drought does not have to continue, as can clearly be observed in panel b) below for the year 2002. Furthermore, for the year 2003, the total length based version shows a very short period of drought. This is contrary to what has been observed within the region, e.g.

Scott et al. (2004 and 2006) have mentioned the occurrence of a long-term drought during this period, which is properly identified with the newly proposed method as presented in the manuscript.

As much as we like the suggestion raised by the reviewer to account for the total length of a dry period as part of the identification method, we have not found the right method on how to implement such an approach in a simple straightforward manner to make it part of the current manuscript. We therefore believe this lies beyond the current scope and decided to only present the results obtained with the method as presented in this work.

Concerning the second point to investigate how long a period of zero flow can occur before a "drought" starts, we will add this information to Section 3.2. We do not agree with the reviewer force droughts to start in the flow season, instead Section 3.2 states:

"*The new method enables identification of drought for days with a small consecutive drought/dry flow number, which are generally observed at the end of the NAM season. Therefore, for these periods, less abrupt changes in the identified discharge percentile are observed for the new method presented here (see also Fig. 4b). The fact that the observed hydrological drought continues after precipitation ceases results in a considerable increase in the total number of drought days for 2002--2005 and 2009--2011 ( Fig.4c).*"

*References:*
*Scott, R. L., Edwards, E. A., Shuttleworth, W. J., E., H. T., Watts, C., and Goodrich, D. C., 2004: Interannual and seasonal variation in fluxes of water and carbon dioxide from a riparian woodland ecosystem, Agric. For. Meteorol., 122, 65 – 84, doi: 10.1016/j.agrformet.2003.09.001,*
*Scott, R. L., Huxman, T.E., Williams, D.G. and Goodrich, D.C., 2006: Ecohydrological impacts of woody-plant encroachment: seasonal patterns of water and carbon dioxide exchange within a semiarid riparian environment. Global Change Biology, 12: 311-324. doi: 10.1111/j.1365-2486.2005.01093.x*

[Figure]

2. I encourage the authors to be more careful with the use of the wording "hydrological extremes" throughout the manuscript. They mostly look at **anomalies**, not extremes. This is an important distinction to make upfront (especially with regard to the communication of the results to a broader audience), especially because some of the "extremes" are not "as extreme" (see next comment)

**We have altered the working throughout the manuscript which now indicates anomalies and not extremes. We agree with the reviewer that anomalies better grasp the content presented in the current work.**

3. Figure 6 (upper panel) reveals that only a very small discharge event (only visible when completely zoomed-in) triggers the identification of a wet period. This period then last for a
30 days as the authors apply a 30-day moving average. I agree that this one-day <0.1 mm discharge event can be abnormal, but to classify this entire period as wet might not be the most suitable.
   a. The authors should discuss this, especially in the context of the term "hydrological extremes" used throughout the manuscript.

**This is a great point. We would also like to refer to our general response given above. It is indeed correct that a small runoff event can cause a longer duration wet period. At first side this might feel incorrect. However, the occurrence of these types of events during the Monsoon season is very rare, as can also be observed from the upper panel of Figure 6. The first small flow peak for the year 2003 did not cause a wet spell anomaly as this flow event happened during July, where historically sufficient flow events have been observed. Only for the very early flow event in early June 2007 did this indeed caused a severe wet spell anomaly, as flow events in late spring are very rare. It should be noted thought that from a hydrological perspective, for semi-arid domains like the San Pedro the occurrence of a small runoff event this early in the season also indicates wetting of the surrounding hillslopes and riparian zone (in order to the cause the flow event), which subsequently stay wetter for a longer period (as compared to a few days for discharge). As such, the "wet" anomaly that occurs for a longer time period, this early in the season basically indicates wet the rootzone conditions (including the channel bed). Strictly speaking this is a wet spell in soil moisture conditions instead of a hydrological wet spell, but for zero flow conditions, the focus is basically on the state of the channel bed. As the channel bed and its riparian zone have a strong ecological functioning (as these are the predominant locations with year-round vegetation and tree cover), we believe that these values higher runoff percentiles are indicative of the state of this system. As stated in our general response the updated version does a much more thorough job in highlighting this aspect.**
**Furthermore, we agree with the reviewer that this is important to mention and have added some additional text to the discussion:**
*"It should be noted that for situations where there is generally no flow, a small flow event can result in a sudden increase of the discharge percentile, as also shown in Fig. 6 for the early summer of 2007. This can result in a wet spell anomaly, which due to a 30-day MA window size, can last for 30 days even though the total flow amount during this period was very small. As the channel flow ceases, the discharge percentile becomes more indicative for the moisture state of the channel bed, which can be expected to be relatively wet due to the strong impact of transmission losses. "*

b. In addition, droughts might develop more gradually then wet periods. Could the authors comment on whether the use of a 30-day moving average window is equally suitable to both drought and wet period?
**We agree with the reviewer that this is drought indeed evolves more gradually then a wet spell. This is also observed is Figs. 4-6 for a constant 30-day MA window. As indicated in point a), we feel that a 30-day period not necessarily will lead to incorrect analyses, as the wet spell for a dry period are derived from runoff observations can be seen as a proxy for the moisture state of the channel bed.**

4. I do not agree that one should definitely correct for human induced step changes and trends for drought analyses (Section 3.3). From an instream perspective, human induced changes and modifications in drought characteristics can be of critical importance as well. In addition, the proposed

method of recalculating drought characteristics for different reference periods (before and after change points, where the change point might vary depending on the river), might hinder a fair comparison of drought characteristics over space and time, as each time a different reference period is used. So, in case the authors want to correct for trends and step changes, it should be better justified why (in Section 3.3), and the consequences should be discussed.

**We thank the reviewer to raise this issue, as we have had multiple discussion on whether to incorporate it. And if so, how? We believe humans induced step changes can potentially have strong impact on the runoff response as has been shown in literature (e.g. Villarini and Smith, 2010; Sadri et al., 2016). If these cause a consequent lowering of flow amounts or discharge maxima, this would result in a sharp change in hydrological extreme characteristics before and after the step change. Similarly, this also holds for longer term changes (as were shown in Figure 2). Therefore, we decided to implement these corrections.**

**The three figures below show the impact of 1) not performing any trend or step corrections, 2) only detrend, 3) only perform step correction, as compared to Figure 6) of the manuscript. These results show that for station 3 (corresponding to Fig. 2):**
1. **Not performing any trend or step corrections would lead to a drying out and system that is almost continuously in drought at the end of the timeseries. We feel that this would only reflect the increased uptake of water by humans throughout time and the impact of sudden modification, while it is the interest of the paper to understand the behavior of hydrological anomalies and how these vary between location.**
2. **Detrending only would not account of the sudden in/decrease of the hydrological response, which can result in a large modification of the data, which for station 3 leads to an under estimation for the discharge percentile. As we indicate in 2.2 there was a downward shift detected in the late 1950's. Detrending without accounting for this shift will raise the effective runoff values during the later period too much, resulting in an underestimation of the hydrological percentile as shown below.**
3. **Performing only the shift identification causes similar problems as in point 1.**

**No shift year and no detrend**

[Figure]

**No shift year, have detrend**

[Figure]

**Correct for shifts year and no detrend**

[Figure]

**We therefore decided to alter the beginning of section 3.3 into:**

**"*As indicated in the Introduction, many of the long-term discharge observations available are impacted by human influences, which can cause both sudden shifts and continuous changes in the observed discharge series (Easterling & Peterson, 1995; Easterling et al 1996; Menne & Williams, 2005). Although, the occurrence of these type of changes was rare for the San Pedro, it was observed for a few locations including station 3 (see also Fig. 2), where both a gradual decrease and shift in the late fifties was observed (see also Fig. 2). By not identifying and correcting for trend and step changes, results would show a drying out of the system, with a wet spell being the predominant condition during the first twenty years, and a drought being dominant during the last twenty years. Although, technically this is correct, we believe this would only reflect the impact of human behavior with increased amount in water uptake throughout time and added modifications throughout the catchment. As stated before, it is the interest of the current work to understand the behavior of hydrological anomalies and how these vary between locations. To be able to assess these, the identification and correction of shifts and trends is necessary, which subsequently allows for the identification and analyses of anomalies within the hydrological system (e.g. Villarini & Smith, 2010, Sadri et al. 2016).*"**

5. I like the consistency of the proposed method, i.e., the 20th percentile is exceeded 20% of the time. However, later on, the authors apply different pooling approaches. Did the authors test how these approaches modify the amount of time anomalies occur for each catchment and DOY?

**This is an interesting question that we did not assess before. Below you'll find two figures that show the average fraction a given method is in drought and the total duration for Station no. 1, 11 and 15. As can be observed, the old method and method proposed in this manuscript without pooling show an average fraction of ~ 18% in drought. The reason why this is slightly lower than the 20% is the fact that for the combined method, the consecutive drought day number distribution percentiles are calculated the distribution of observed dry days (step 2 in both Sections 3.2 and 3.3), while the actual percentile is calculated on the basis of the combined consecutive drought/dry day number (step 3 in both Sections 3.2 and 3.3). As this latter number tends to be higher for a given day, the corresponding discharge percentile is lower.**
**For the pooling method, these results nicely show that the fraction of drought increases both with interevent time as well as the maximum upper percentile value. As we feel that this paper contains enough figures, we decided not to add the figure below to the manuscript but instead add an extra line to Section 4.3 stating:**

**"As a result, the total fraction of time that a hydrological drought is observed increases slightly with both interevent time and upper drought percentile threshold $\mathrm{Pu}$ (not shown here)."**

[Figure]

Minor comments:
The manuscript requires some more thorough editing. Suggestions (and some other points) in the minor comments below.
**We very much appreciate the suggestions raised by the reviewer, these have improved the readability of the manuscript.**

- Title: a major part of the paper deals with modifying the combined TLM and CDPM. I think this should somehow be reflected in the title.
**We will include this aspect in a the newly updated title**

- Line 3: regularly occurring no flow conditions instead of "no flow conditions"
**Done!**

- Line 5: The wording "hydrological extremes". I would be very careful with this wording throughout the manuscript. You mainly identify anomalies, not extremes.
**We agree with the reviewer and have changed "hydrological extremes" into "hydrological anomalies", throughout.**

- Line 16: the abstract does not mention anything specific about the impact of elevation although it is in the title. Maybe it can be added here.
**We thank the reviewer for pointing that out and have extended the original sentence:**
**"This specifically holds for catchments with no perennial flow"**
**Into:**
**"This specifically holds for catchments with no perennial flow, which are situated at lower elevation and do not receive additional inflow from regions upslope."**

- Line 17: correct the spelling of "drought and wet spells patterns"
**Done.**

- Line 18: wet spell**s** and drought**s** instead of "wet spell and drought are rare".
**Done.**

- Introduction: I strongly encourage the authors to review some more recent literature, especially about the application of the combined method between 2012 and now.
**We will add this throughout the manuscript.**

- Introduction: be more careful in your referencing, e.g.: Line 25: The sentence about the costs of drought with a reference to Dai (2011) is also in the Van Huijgenvoort paper. However, they refer to a different contribution of Dai (2011).
**Thanks for noticing this! We used indeed the wrong reference to our bibtex file and have updated this.**

- Line 38: "severe dry spells impacting the land surface". You could be clearer here.
**We changed this into:**
**"showed increased amounts of intensive dry spells in recent decades, impacting land-atmosphere interactions and the hydrological response.**

- Line 38: Where does "This" refer to?
**We changed this into:**
**"These dry spells can cause more extreme drought"**

- Line 41: if you focus on both dry and wet spells, it would be good to give them equal amount of attention. Now most focus on droughts.
**We made sure to increase the weight of wet spells throughout the paper. For instance, we have added references that focus on this. Furthermore, the results presented in this work focus both on both drought and wet spells. Last, in section 4.2 we have highlighted that the original approach of Van Huijgevoort only focused on drought, while the newly combined method presented here is able to identify wet spells during the dry season (as a proxy for wet channel bed root zone conditions).**

- Line 43: remove "(" after e.g.
**Done.**

- Line 50: You could add to this argument that some studies kick-out stations with regularly, occurring zero flows. (I am guilty as well; Tijdeman et al., 2016).
**We added: or removed locations where zero flow conditions occur regular (Tijdeman et al., 2016)"**

- Line 51: "incorrect higher threshold". Why is a higher threshold incorrect? What is a correct threshold?
**Our original motivation was related to our original assumption of a drought occurring about 20% of the time. However, we agree with the reviewer that this is not needed here and have removed the word "incorrect" here.**

- Line 54: if true, the authors could add that the CDPM was developed for precipitation.
**Done.**

- Line 56: The main problem of the method was the difficulty to characterize the transition from flow to low-flow periods, or? Not the identification of the start.
**Correct. We changed this sentence into:**
**"this method has difficulty to characterize the transition from flow to low-flow periods for regions with a strong seasonal precipitation signal"**

- Line 61: elevation is not a direct control. In this study, elevation encompasses a variety of different controls such related to snow, the occurrence of groundwater, and the occurrence of local rainfall events.
**This part is removed within the new version of the manuscript.**

- Line 68: "To limit the impact of human influences" I would remove this from the sentence as it reads as one of the goals from the study.
**Done.**

- Line 82: suggest to remove "detailed"
**Done.**

- Line 83: suggest to remove "detailed"
**Done.**

- Line 86: "as explained in the introduction" If you explain something in the introduction, you do not have to repeat.
**We removed the first sentence from the manuscript**

- Line 92: "300-350 mm of precipitation" ☐☐per year?
**We added "per year" to this sentence.**

- Line 110: Be consistent. You use both stream flow and streamflow.
**Done. We removed "stream flow" from the paper and instead used "streamflow" throughout.**

- Line 118: spell instead of "spells"
**Done.**

- Line 134: you could also refer to the threshold level method here.
**Done. We have added "and Threshold Level Method (e.g. Yevjevich, 1967; Tallaksen et al 1997, 2009; Fleig et al. 2006)"**

- Line 135: you could explain why these methods do not work well in these domains.
**We have added: ", as they cannot distinguish between a normal and drought situation in case of zero flow."**

- Line 138: More careful phrasing: I doubt that this is the most widely used method, especially not for wet "hydrological extremes".
**We changed this sentence into:**
**"The TLM is widely used to identify hydrological anomalies for rivers with perennial runoff"**

- Line 139-141: More careful phrasing needed as this is not necessarily true. Some (of the mentioned) studies methods use daily flow values to define a monthly or annual threshold (e.g., fixed threshold).
**Will make sure to reference this correctly.**

- Line 143: All referenced studies are drought studies. Any example of the use of the TLM to identify wet spells?
**We added the reference to Zhoa et al. (2009) and Garner et al. (2015).**

- Line 143-144: the threshold level method can never distinguish between a dry period and drought. The problem with regularly occurring zero flows:
o Droughts are indicated for all zero flows or for none of the zero flows, depending how you define your quantiles.

o It is quite complicated to rank zero flow events, i.e., how do you rank five zeros (see Stagge et al. (2015) how they deal with zero precipitation in the SPI).

o These problems should be presented more clearly (either here or in the introduction), as you are going to solve them later.

**We thank the reviewer for mentioning the paper of Stagge et al. (2015) as they present an interesting procedure to solve the "zero" precipitation problem. As stated above in our comments to major point 3, we believe it is possible to differentiate between different zeros as a higher consecutive drought/dry day number is indicative of dryer channel bed conditions. We believe the method presented here tries to properly rank these zero flow events, although will be the first to acknowledge there are limitations to our approach (see also our response to your main comment 1). In order to explain the problem with the zero flow conditions a bit better, we have added the following to Section 3.1:**

**"Unfortunately, for locations with zero flow conditions occuring a fraction *P* of the time, the minimum flow percentile observed using the TLM approach would be *P* as the method is not able to distinguish between different zero flow conditions. It should be noted that a similar situation is also observed for consecutive periods with precipitation (Stagge et al 2015)."**

- Line 162: I would describe the rescaling method more clearly.

**Done. These lines now state:**
**"Its corresponding dry/drought fraction $F_{dry/drought}$ is obtained by subtracting the given cdf value from one, as higher consecutive drought/dry day numbers indicate dryer conditions. The final discharge percentile is then obtained by multiplying this dry/drought fraction $F_{dry/drought}$ with the fraction of time zero flow conditions occur for a given day $P_{dry}$. This rescaling will ensure that a given zero runoff day will receive a discharge percentile that will always be equal or below $P_{dry}$.**

- Line 164: ">20%" this is not a wet spell.

**This was a type. We changed this to ">80%".**

- Line 166: "Major advancement" □□Any references to studies that applied this method would be nice her. Here, or in the introduction.

**We added the following references here as well as in the introduction: Breyer et al. (2018) and Heinke et al (2019).**

- Line 204: why 25 and 75? Why not 20 and 80 (your drought and wet spell thresholds).

**We will do this but expect no serious changes in the overall results.**

- Line 182-187: Maybe use bullet points here as well. This would highlight the modifications you made.

**Done.**

- In general, I would strongly encourage the authors to use some variables and equations in both the new and modified method description (as is done in the prev. paper, where it increased the clarity).

**We have added this where we thought was helpful. The reason why we did not do this in the first submission, while describing the original approach is that the current paper only provides a brief summery. For a full description the reader can read the original manuscript by Van Huijgevoort et al. (2012). However, we agree with the reviewer that this sometimes can be helpful, so added this in Section 3.1 and 3.2.**

- Section 3.3. A reference to Sadri et al., 2016 (HESS) would fit nice here (uses same kind of tests to identify human modifications of low flow).

**Done, see also major point 3.**

- Line 200: "as indicated in the introduction …" -> repetitive and not needed.
**These lines will be removed from the text.**

- Line 202: "is necessary to identify and correct for these impacts" do not agree (see major comment (3). Please elaborate why.
**We refer to our response to major comment 3.**

- Line 224: "The previous section presented …" ⬜⬜suggest to delete.
**Done.**

One missing first result section would be the evaluation of the newly proposed method / comparison between the new and old method. (4.1). Now, this is more or less done in the methods section. However, I believe this better fits the results.
**We have added a new section 4.1.**

- Line 231: The fact that you apply a moving average on your data should be presented earlier (before the introduction of the combined method).
**Step 1 in Section 3.1 was added which reads:**
**"A backward looking moving average filter with given window size is used to smooths observations"**

- Line 233-234: Did the authors recalculate the threshold after applying a different moving average interval?
**Yes. We added the following to these lines: ", with the thresholds being recalculated for each window size".**

- Line 235: Introduce abbreviation "PDSI"
**Done.**

- Define duration in the methods and be consistent. Is this the total time in drought for a certain year (as x-axis Figure 4c) or the length of the drought / dry spell (e.g. Figure 8).
**Done.**

- There are many unnecessary repetitions at the start of each new section (e.g., the first five lines of 4.3). I would suggest deleting these to improve readability.
**We removed these sections as suggested by the reviewer.**

- How are probabilities in Figure 8, 9, 10 etc. calculated. And can the authors confirm that the dips in probabilities at the left and right side of Figure 8b,e are actually there? And not an artefact of any kind of smoothing that does not take into account observation before and after the plotted period?
**This is indeed an important point. We used the "density" function in R programming language package stats v3.6.1. As the reviewer pointed out, there is an edge effect of the density function as mentioned above. 1000 sets of 55 random numbers are uniformly drawn from 1 to 365, the density function is applied to each set and the graph above shows the average of these 1000 density fits. In order to correct for this we took extended the observational data below day one with the data at the end of the year, while we extended the data beyond day 365 with those observed for the beginning of the year. We believe that this resolved the issue of the edge effect.**

[Figure]

- Line 251-255: This is repetition and could be deleted.

**Done. Note that we added a new section 4.1 as suggested by the reviewer.**

- Line 257: As indicated in Section 2.2 … (repetition).

**This line was removed. The first line of this section now reads:**

**"For a representative station from each runoff category (see Section 2.2), the observations, discharge percentiles and identified drought and wet spells are given in Figure 6 for the years 2002--2007."**

- Line 258: spells instead of "spell"

**Done.**

- Line 258: I could not find a reference to Figure 5 before (and after) you introduce Figure 6.

**In line with the suggestion raised by the reviewer, we have created a new section 4.1.**

- Line 268: Discharge instead of "Moisture"

**Done.**

- Line 285: For both stations?

**We changed this sentence into:**

**Drought and wet spells for the two locations with baseflow (Stations 9 and 3) are much more fragmented throughout time, although longer duration drought and wet spells lasting multiple seasons can be observed for both stations (e.g. around 2003 and 2010).**

- Line 301: For the representative station or for all stations?

**We added "for all stations" to this line.**

- Line 304: I would not call a 100 days "very short"

**We changed this into "only a few months".**

- Line 346-350: This is all repetition, which the authors could consider to remove.

**Done.**

- Line 355-356: to much brackets, especially to much ")" …

**We changed this into:**

**"Overall, the biggest impact of pooling is observed for the maximum upper drought percentile threshold ($P_u$=50% for top right panel Fig. 9) or minimum lower wet spell percentile threshold ($P_l$=50% for bottom right panel Fig. 9).**

- Line 368: Use of wording "hydrological extremes" (see earlier comment)
**This was changed throughout the manuscript.**

- Results general: there is some discussion in the results (e.g., 295-300). I would suggest the authors to more strictly separate.
**We moved some of this to the discussion.**

- Discussion general: too much repletion of the results and too less interpretation. For example, various other studies show the effect of pooling and smoothing, and the authors could relate their findings to these works (+ a recommendation which smoothing method to use). In addition, there are also quite some new results in the discussion. I suggest separating.
**We agree with the reviewer that the discussion is indeed too long. The reason why we added these new results here is that we felt that we should use the discussion to bring the different results together (as shown by these additional two figures), as well as create the additional discussion. We have moved these figures to a newly created result section 4.5. Note that the updated versions provides more in depth information on hydrological anomalies and there impacts within semi-arid regions and we address some of these aspect in the discussion.**

- Line 408: "impact of elevation and flow conditions." Elevation does not have a direct impact (and "flow conditions" is a bit of a vague term).
**We do agree with reviewer on the term elevation, as indicated above and added to the introduction, elevation comprises the joint impact of snow, a reduction in temperatures and the impact of groundwater, within the southwest US. However, we did agree with the reviewer that flow conditions was a vague term and therefore suggest to use baseflow instead. This has been updated throughout the manuscript.**

- Line 426: "dominant driver" of what?
**This was remove from the text as we remove figure 13.**

- Line 432: "short duration hydrological extremes" … again not extreme. Anomaly.
**Done.**

- Figure 2: You might add the trend from 1960-onward to the legend to show that it is less or absent.
**Done. We added: "Note, for the period 1960-onward no statistically significant trend can be observed." to the legend.**

- Figure 2: Location nr. 2 has only data from 1967 –onward (according to table 1). Location 3 or error in table 1?
**This was indeed Location 3. We updated the caption accordingly.**

- Figure 3: I have difficulties to compare the timeseries in the panel plots. It might be better to separate them? And maybe use a log-scale, so you can see higher flows as well?
**Done. We have moved the insets to the upper row and transformed the y-axis into logarithmic scale.**

- Figure 2,3,4,5: be consistent. You now use discharge (mm) vs. runoff (mm) vs. discharge (m3/sec) vs. moving average discharged(?) in Figure 5.
**Thank you for picking that up. We have changed this into:**
**Figure 3: We have changed the y-axis into "Discharge (mm day^-1)"**
**Figure 4a: We have changed the y-axis into "Discharge (mm day^-1)"**
**Figure 5a: We have changed the y-axis into "Discharge (mm day^-1)"**

- Figure 4c, x-axis: duration or days in drought?

**We have changed the x-axis in "Time" and the y-axis of into "Duration".**

- Figure 4, caption: remove ")"
**Done.**

- Figure 7: I would not use green for wet spells (subjective opinion). Maybe light blue as is done in the USGS WaterWatch? https://waterwatch.usgs.gov/
**Done. We changed this to the color light blue.**

- Figure 8: I would separate the probability plots from the main plots, as these are not really related.
**Done. We created a 6 panel figure without the insets.**

- Figure 8: do the lower probabilities on the right and left side of the probability plot indicate a lower probability of occurrence (e.g., left of Fig. e)? Or is this some kind of a smoothing effect (due to a possible absence of points at the start and the end of the year).
**This is the smoothing effect.**

- Figure 8: modify the plotting area to fit all observations.
**Done. We have extended the y-axis in towards 450.**

- Figure 8-11: What is the difference between the probability (8-9) and density (10-11)?
**There is not specific difference, so we changed the word "Probability" in Figures 8 and 9 into "Density".**

- Figure 11: Why a MA period of 1 year? If you apply a one-year MA, you probably remove all zero values and are comparing results for the TLM.
**Correct. When we discuss Figure 10 in Section 4.3 we state: "For a one and five year windows, zero flow conditions are rare and for these results the TLM approach was used."**
**The 'general response" above addresses the reason why we also included these long MA-windows. We will ensure that this is properly added to the updated version of the manuscript.**

- Figure 12: Any pooling applied here?
**No. We added "without pooling" to the legend.**

- Figure 13: For similar reasons: why a 1-year MA average window.
**The 'general response" above addresses the reason why we also included these long MA-windows. We will ensure that this is properly added to the updated version of the manuscript.**

---

## Author Comment (AC2) · 11 Feb 2020

***bold indicates our response.

Reviewer 3

**Our general response**
The original manuscript submission did a poor job delivering a number of messages and providing additional details. We have tried to address these in an updated version of the manuscript. The main comments raised by all three reviewers can be summarized in the following three major concerns:

1. The manuscript lacks a strong motivation why one would be interested to focus on drought and wet spells within a semi-arid environment that is predominantly dry throughout the year.
2. A good description of the newly presented combined identification method including a logical figure.
3. An incorrect focus where ephemeral rivers were compared with perennial rivers, without providing the necessary relationships.

We have tried to address these major issues in a new version of the manuscript and we believe that this has strongly improved the manuscript readability and clarifies the above mentioned topics. This original submission was poorly presented and we would like to thank the reviewers for identifying this. However, we also believe that the updated version provides considerable interesting insights into the occurrence and variability of semi-arid hydrological anomalies that strongly reflect the ecohydrological functioning of the channel bed and riparian zone, which we believe are of interest to readers of HESS.

**Major concern 1:**

**To the introduction we have added the following:**

**"This combined procedure was able to identify hydrological anomalies specifically within transitional regions, where zero flow conditions are common but not the standard. However, as the current work will show, this method has difficulty correctly identifying drought and wet spell occurrences within ephemeral rivers within semi-arid regions with a strong seasonal precipitation signal (e.g. Monsoon). The strength and occurrence of seasonal runoff within these rivers strongly impacts groundwater recharge, and the ecohydrological state of the channel bed and riparian zone (Goodrich et al., 2004; Scott et al., 2008). As a result, these reaches become hotspots for biodiversity, especially during the dry season (e.g. Moreno-de las Heras et al., 2012; Cleverly et al., 2016). Correct identification of a hydrological anomaly is therefore important. As a result, this paper presents an updated version of the combined approach originally developed by Van Huijgevoort et al. (2012), improving the identification and continuation of a drought or wet spell, when transitioning from the wet into the dry season. This newly combined approach provides information on the hydrological status of the river including the occurrence of anamolies. For semi-arid ephemeral channels this information serves as a proxy for the moisture state of the channel bed and riparian zone, and its ecohydrological functioning (Scott et al., 2008; Moreno de las Heras et al., 2012). Specifically, this paper focuses on the semi-arid southwest US, where considerable amounts of long-term observations are available, which allows for a detailed analysis."**

**To Section 2.1 we have added the following:**

**"Within the San Pedro, upslope headwaters have low permeability bedrock relatively close to the surface, resulting in infiltrating rainfall and snow melt quickly to reach the groundwater system and river network (Fan et al., 2007; Kampf et al., 2016). These perennial streams become ephemeral downslope through transmission losses from evaporation and infiltration into the dryer ambient subsurface (Cataldo et al., 2010; Blasch et al., 2013). Within the semi-arid San Pedro basin, stream infiltration has been shown to account for 10-40% of total groundwater recharge (Goodrich et al., 2004). However, for the majority of precipitation generating flow events the depth of infiltration is relatively shallow, with only the biggest events infiltrating deeper (>1 meter below the channel bed). Transmission losses therefore strongly impact root zone moisture availability. Lower elevation riverbeds and their**

surrounding areas are therefore favorable for biodiversity (e.g. Moreno-de las Heras et al., 2012; Cleverly et al., 2016)."

**To Section 3.2 we have added the following:**

"As indicated in the introduction, flow events replenish rootzone moisture below the channel bed, which is used by the riparian zone vegetation during the dry season for transpiration. Therefore, the occurrence of a drought or wet spell during the positive flow season, directly impacts moisture availability afterwards. As such, it is important that a given hydrological state continuous from the wet into the dry season. For a hydrological drought to prolong from the wet into the dry season, drought should also be identified at lower consecutive drought/dry day numbers, which currently is impossible."

**To the discussion we added the following:**

"It should be noted that for situations where there is generally no flow, a small flow event can result in a sudden increase of the discharge percentile, as shown in Fig. 6 for the early summer of 2007. This can result in a wet spell anomaly, which due to a 30-day MA window size, can last for 30 days even though the total flow amount during this period was very small. As the channel flow ceases, the discharge percentile becomes more indicative for the moisture state of the channel bed, which can be expected to be relatively wet due to the strong impact of transmission losses."

**Where we felt needed, we refer to these sections throughout the rest of the manuscript. By providing these details we believe the motivation and focus behind this work become clear to the reader.**

**Major concern 2:**

**To address this point, in Section 3.2 we have tried to identify the current limitation of the original combined method of Van Huijgevoort et al. (2012). Furthermore, in our original submission we did not highlight that this method was only developed for the identification of drought. Also, the updated version provides now a link to point 1 raised above, to highlight that a given hydrological state as well as the occurrent of a drought and wet spell, here reflects that available root zone moisture.**

**Therefore, Section 3.2 was completely rewritten. The first part now contains the following:**

"Even though the combined method proposed by Van Huijgevoort et al. (2012) has been a major advancement to identify hydrological anomalies within transition regions (e.g. Breyer et al., 2018; Heinke et al., 2019), the method has three potential issues. First, the final discharge percentile distribution for a given day of the year does not have to be perfectly uniform between 0–100. This is the result of combining the original TLM, where discharge percentiles are obtained for a given day, with the CDPM, which defines the cumulative density function (cdf) on the bases of all dry days. Even though consecutive drought/dry numbers are rescaled (see step 3 of Section 3.1), this does not guarantee a perfect uniform distribution between 0 and 100 for a given day.
Second, during zero flow situations, drought only occurs in case a given day has a high consecutive drought/dry day number. For the San Pedro these occur in spring at the end of the dry season. Therefore, in case a drought starts during the positive flow season in summer, it will generally stop identifying the drought once flow ceases. If zero flow conditions continue to occur, the combined method will then identify a drought again later in the dry season in spring. As such, the original combined method therefore identifies two droughts (i.e. one in summer during the NAM and one in spring after a prolonged period without flow, see also Section 4.1). As indicated in the introduction, flow events replenish rootzone moisture below the channel bed, which is used by the riparian zone vegetation during the dry season for transpiration. Therefore, the occurrence of a drought or wet spell during the positive flow season, directly impacts moisture availability afterwards. As such, it is important that a given

hydrological state continuous from the wet into the dry season. In order for a hydrological drought to prolong from the wet into the dry season, drought should also be identified at lower consecutive drought/dry day numbers, which currently is impossible.
Third, originally the method was not developed to identify wet spells for zero flow conditions, as these would always occur during situations of low consecutive drought/dry day number. For the San Pedro, this would result in a wet spell to occur immediately at the start of the dry season, even if a drought was observed during the positive flow season (see also Section 4.1)."

We also extended the explanation of the new combined procedure presented, following an approach similar to the one presented in the original work of Van Huijgevoort et al (2012) making use of both mathematical symbols and enumerated lists. Here we use a figure to indicate the different steps.

Furthermore, in line with suggestion raised by the reviewer, we moved the figure that showed the difference between the old and new combined method to a newly created Section 4.1, which contains the following:
"4.1 Comparison of the original and modified combined method
Figure 5 shows the difference in identified discharge percentile between the combined method as proposed by Van Huijgevoort et al. (2012) (old method) and the modified combined method proposed here (new method). As indicated in Section 3.2 the old method was unable to identify drought during zero flow conditions with a low consecutive drought/dry day number. This can clearly be observed from Fig. 5b where these days, that generally occur at the beginning of the dry season during the fall, show high corresponding discharge percentiles. Furthermore, for the years 2004–2005 and 2009–2010, the drought observed during the positive flow season, first ceases and return the following spring at the end of the dry season for high consecutive drought/dry day number. This situation does not occur for the new method presented here, where the consecutive drought/dry day number of a given zero flow day is compared with its number observed during other years. As such, discharge percentiles in the fall can be small enough to indicate a drought. As a result, for both 2004–2005 and 2009–2010 the drought continued from the wet season into the fall, as an indication of the moisture state of the riparian zone. The fact that the observed hydrological drought continues after precipitation ceases results in a considerable increase in the total number of drought days for 2002–2005 and 2009–2011 (Fig. 5c).
Section 3.2 also mentioned that, the original method of Van Huijgevoort et al. (2012) would lead to high discharge percentiles at low consecutive drought/dry day number and as such was not specifically developed for this. For the fall, Fig. 5b shows for station 11 this would indicate the occurrence of a wet spell during the fall of almost each year, irrespective of the strength of the NAM. The newly combined method presented here, does not show this strong dependency. In stead, high discharge percentiles in the fall only occur during years where already during the NAM a wet spell was observed (i.e. 2000–2001 and 2006–2007). During the other years discharge percentiles in the fall do not indicate the occurrence of a wet spell."

We believe these changes have improved the description of the new combined procedure presented here and its applicability to identify hydrological anomalies within semi-arid ephemeral channels.

Major concerning 3:
As the current work focuses on hydrological anomalies within semi-arid environments and more specifically the San Pedro basin, we feel that only focusing on the ephemeral channels as suggested by reviewer 2 would not provide a complete picture of hydrological anomalies occurring throughout this region. Instead we assess their occurrence of three typical cases: 1) dry lands channel beds that only show runoff from intense precipitation events during the NAM and from remnants from hurricanes in the fall (NB), 2) the upland regions that have perennially flow conditions (BU), and 3) low land regions with perennially flowing conditions received from baseflow upslope (BD). To introduce this we added the following to the introduction:

"Results obtained for ephemeral channels as derived using the new combined procedure presented here, will be compared with observations from perennial rivers at higher elevation

as well as for downslope locations receiving continuous flow from upslope. By analyzing the occurrence of drought and wet spell anomalies across these locations, this work will provide a detailed overview of their occurrence and variability within the San Pedro, as well as the role of climate and local geographical location."

For case a) dry lands channels, the occurrence of a single event can generate a wet spell even after runoff ceases. This might be counterintuitive, as indicated by all three reviewers, as "How can zero flow conditions correspond to a wet spell".  We hope that the comments provided by major point 1 and 2 as given above, have resolved this aspect.

Reviewer 2 indicated that one option would be to only focus on case a). As we did a poor job describing the different system, we understand this suggestion. However, we hope by correctly adding these details throughout the paper, it becomes clear why it is important also to address the uplands.

For case b) the hydrological response of these the upslope regions is much more similar to temperate environment. Hydrological drought and wet spells have a direct link to the amount of water available in the river network. Furthermore, as flow conditions are always positive, we added the following to Section 3.2:

"for locations within continuously flowing condition (BU and BD category) the TLM approach was used solely."

For case c), at shorter time scales (30-day moving average window), observed hydrological drought and wet spell characteristics resemble those observed for the upland reaches, since during the dry season all water originates from these uplands. However, also in these environments, transmission losses form an important source of moisture for the surrounding riparian zone. Since these channels transport the majority of their water during the NAM, when focusing on a longer timescale (e.g. MA-window of 1 year), their hydrological drought and wet spell characteristics resemble more case a) the NB category. As such, at these timescales a hydrological anomaly is more representative for the state of the riparian zone. Again, the paper did a poor job in explaining this.

Because of this difference in behavior across MA-window scales, we felt it was important to assess this. Therefore, we presented two figures for 30-day MA windows (Figs 6-8), various MA-window sizes (Fig. 10) and a one-year MA-window size (Figs. 9 and 11). However, we did not properly motivate these choices and have tried to include this in the updated version of the paper, by addressing these in both the Results section as well in within the Discussion. The latter now states:
"The mean duration of a wet spell last longer for upslope domains (BU) as compared to the lower elevation categories (BD and NB). For the upslope regions, shallow groundwater is expected to have a stronger control on the observed amount of baseflow for a longer period of time, increasing the mean duration (see also Section 4.2)."

 Besides addressing these major concerns we have:

- **Changed the title as suggested by the reviewers**
- **Include more up to date references in introduction and section 3**
- **We propose to alter the setup of the manuscript by creating a new paragraph 4.1, before the original sections 4.1-4-3 which will become section 4.2-4.4.**
- **For the one-year MA-window analyses in Figures 11 and 13, we have added the details of why we feel this is interesting both in the Introduction as well as in Section 3.4 and in the Discussion. We will also highlight that this situation effectively corresponds to applying the TLM only.**
- **To decrease the length of the discussion and not to present new results we have moved the figures of the discussion of the original manuscript to a newly created section 4.5.**
- **Adjust the original figures where suggested.**

**We are convinced that by resolving these concerns and addressing the various issues raised by the reviewers the quality of the manuscript has improved consirably.**

This manuscript deals with a new methodology for assessing hydrological drought with a variable threshold approach in semi-arid catchments and its application on case study catchments in Arizona.

**We would like to thank reviewer 3 for the constructive comments. We realize that reviewer 3 was very critical with respect to the quality of the original submission. Based on the suggestion raised by the reviewer we believe we have improved the focus and quality of this work considerably.**

**Main comments**
The comments will clearly overlap with those of the other referees, emphasizing the main issues found in the manuscript. But before that, it has to be noted that the study is reasonably well conducted and the manuscript generally well written and organised.

However, several major issues can be identified:
1 The study extends some previous work on a generic method for defining a hydrological drought with a variable threshold (Van Huijgevoort et al., 2012), a method that would be applicable to semi-arid catchments with long periods of zero flows. First, the methodological extension presented L165-L198 is not clearly described, and would deserve some more didactic illustration and/or pseudo-algorithm.

**We agree with the reviewer that the original manuscript did not do a good job in explaining the updated method. We refer to major concern number 2 as defined above. We have increased the description of the methon, making it inline with the original method of Van Huijgevoort et al (2012) as well as including an additional figure we hope to have addressed this accordingly.**

2. Second, one wonders about the usefulness of proposing an extension of the variable-threshold hydrological drought definition. Indeed, the basic reasoning of using a variable threshold is to detect streamflow lower than usual. When streamflow is usually zero, what is lower than usual? The conceptual limits of the variable threshold method are here clearly reached. Indeed, what is the point of by all means trying to compute/define a streamflow deficit when there is no streamflow? I guess that defining drought in a non-perennial river may be useful, for example for aquatic biodiversity, but streamflow drought is here not relevant. What could be relevant for example in this case is defining an edaphic drought in the river bed to assess the decline of soil moisture during the dry season, and the state of the corresponding habitat for invertebrates.

**The reviewer is indeed correct! We did a very poor job in addressing this aspect in the original manuscript (see also our general response above. We refer to our major concern 1 where this comment was treated explicitly. By updating the different components throughout the manuscript, we feel we have properly addressed this aspect.**

3. Third, and in relation to the above comment, the objective of the study is clearly ambiguous. Indeed, the title does not even mention the proposition of a new methodology. Second, the current title suggests some relationships between catchment characteristics and drought characteristics, and the way results are presented – including the poor map of Figure 1 that don't even show the delineation of catchments – do not allow to even extrapolate results in other basins with similar characteristics.

**Agreed. Though we feel that it is not so much Figure 1 that should address this, but in stead, additional information in the text providing more detailed information about the three different hydrological categories and there locations within the landscape. Furthermore, it should be noted that the title will be altered in line with the suggestion raised by the different reviewers.**

4. Fourth, the interpretation of results in terms of hydrological droughts as computed with the newly proposed method is in my view just that: an interpretation. Indeed, this is when results are presented

that the reader realises that there is no possible assessment of whether the new method is more relevant than another one, because precisely of the questionable relevance of using a variable threshold for zero flows. This method indeed comes down to assess how much the river is not flowing...

As a conclusion, a new methodology to compute hydrological drought – understood as anomalies with respect to a variable threshold – for semi-arid basins has to show a continuity with perennial basins in order to be taken as a serious candidate. This would in my opinion be the only way to assess the underlying computation assumptions. This is what the authors have tried to present, but results are unfortunately unconvincing. And even with such a continuity, I definitely question the overall approach of hydrological drought understood as anomalies when considering non-perennial basins. This study has indeed one merit: to exemplify the irrelevance of considering variable threshold approaches for characterizing hydrological droughts. As an example on the other side of the hydrological spectrum, I am not sure why a slightly-shorter-than-usual severalmonth-long period with a dry riverbed would be named a wet spell for a given basin.

**As indicated, we would like to refer to our general response stated above. We believe this addresses the different concerns raised by the reviewer.**

---

## Author Comment (AC3) · 11 Feb 2020

***bold indicates our response.

Reviewer 2

Our general response

The original manuscript submission did a poor job delivering a number of messages and providing additional details. We have tried to address these in an updated version of the manuscript. The main comments raised by all three reviewers can be summarized in the following three major concerns:

1. The manuscript lacks a strong motivation why one would be interested to focus on drought and wet spells within a semi-arid environment that is predominantly dry throughout the year.
2. A good description of the newly presented combined identification method including a logical figure.
3. An incorrect focus where ephemeral rivers were compared with perennial rivers, without providing the necessary relationships.

We have tried to address these major issues in a new version of the manuscript and we believe that this has strongly improved the manuscript readability and clarifies the above mentioned topics. This original submission was poorly presented and we would like to thank the reviewers for identifying this. However, we also believe that the updated version provides considerable interesting insights into the occurrence and variability of semi-arid hydrological anomalies that strongly reflect the ecohydrological functioning of the channel bed and riparian zone, which we believe are of interest to readers of HESS.

Major concern 1:

**To the introduction we have added the following:**

**"This combined procedure was able to identify hydrological anomalies specifically within transitional regions, where zero flow conditions are common but not the standard. However, as the current work will show, this method has difficulty correctly identifying drought and wet spell occurrences within ephemeral rivers within semi-arid regions with a strong seasonal precipitation signal (e.g. Monsoon). The strength and occurrence of seasonal runoff within these rivers strongly impacts groundwater recharge, and the ecohydrological state of the channel bed and riparian zone (Goodrich et al., 2004; Scott et al., 2008). As a result, these reaches become hotspots for biodiversity, especially during the dry season (e.g. Moreno-de las Heras et al., 2012; Cleverly et al., 2016). Correct identification of a hydrological anomaly is therefore important. As a result, this paper presents an updated version of the combined approach originally developed by Van Huijgevoort et al. (2012), improving the identification and continuation of a drought or wet spell, when transitioning from the wet into the dry season. This newly combined approach provides information on the hydrological status of the river including the occurrence of anamolies. For semi-arid ephemeral channels this information serves as a proxy for the moisture state of the channel bed and riparian zone, and its ecohydrological functioning (Scott et al., 2008; Moreno de las Heras et al., 2012). Specifically, this paper focuses on the semi-arid southwest US, where considerable amounts of long-term observations are available, which allows for a detailed analysis."**

**To Section 2.1 we have added the following:**

**"Within the San Pedro, upslope headwaters have low permeability bedrock relatively close to the surface, resulting in infiltrating rainfall and snow melt quickly to reach the groundwater system and river network (Fan et al., 2007; Kampf et al., 2016). These perennial streams become ephemeral downslope through transmission losses from evaporation and infiltration into the dryer ambient subsurface (Cataldo et al., 2010; Blasch et al., 2013). Within the semi-arid San Pedro basin, stream infiltration has been shown to account for 10-40% of total groundwater recharge (Goodrich et al., 2004). However, for the majority of precipitation generating flow events the depth of infiltration is relatively shallow, with only the biggest events infiltrating deeper (>1 meter below the channel bed). Transmission losses therefore strongly impact root zone moisture availability. Lower elevation riverbeds and their**

surrounding areas are therefore favorable for biodiversity (e.g. Moreno-de las Heras et al., 2012; Cleverly et al., 2016)."

To Section 3.2 we have added the following:

"As indicated in the introduction, flow events replenish rootzone moisture below the channel bed, which is used by the riparian zone vegetation during the dry season for transpiration. Therefore, the occurrence of a drought or wet spell during the positive flow season, directly impacts moisture availability afterwards. As such, it is important that a given hydrological state continuous from the wet into the dry season. For a hydrological drought to prolong from the wet into the dry season, drought should also be identified at lower consecutive drought/dry day numbers, which currently is impossible."

To the discussion we added the following:

"It should be noted that for situations where there is generally no flow, a small flow event can result in a sudden increase of the discharge percentile, as shown in Fig. 6 for the early summer of 2007. This can result in a wet spell anomaly, which due to a 30-day MA window size, can last for 30 days even though the total flow amount during this period was very small. As the channel flow ceases, the discharge percentile becomes more indicative for the moisture state of the channel bed, which can be expected to be relatively wet due to the strong impact of transmission losses."

Where we felt needed, we refer to these sections throughout the rest of the manuscript. By providing these details we believe the motivation and focus behind this work become clear to the reader.

Major concern 2:

To address this point, in Section 3.2 we have tried to identify the current limitation of the original combined method of Van Huijgevoort et al. (2012). Furthermore, in our original submission we did not highlight that this method was only developed for the identification of drought. Also, the updated version provides now a link to point 1 raised above, to highlight that a given hydrological state as well as the occurrent of a drought and wet spell, here reflects that available root zone moisture.

Therefore, Section 3.2 was completely rewritten. The first part now contains the following:

"Even though the combined method proposed by Van Huijgevoort et al. (2012) has been a major advancement to identify hydrological anomalies within transition regions (e.g. Breyer et al., 2018; Heinke et al., 2019), the method has three potential issues. First, the final discharge percentile distribution for a given day of the year does not have to be perfectly uniform between 0–100. This is the result of combining the original TLM, where discharge percentiles are obtained for a given day, with the CDPM, which defines the cumulative density function (cdf) on the bases of all dry days. Even though consecutive drought/dry numbers are rescaled (see step 3 of Section 3.1), this does not guarantee a perfect uniform distribution between 0 and 100 for a given day.
Second, during zero flow situations, drought only occurs in case a given day has a high consecutive drought/dry day number. For the San Pedro these occur in spring at the end of the dry season. Therefore, in case a drought starts during the positive flow season in summer, it will generally stop identifying the drought once flow ceases. If zero flow conditions continue to occur, the combined method will then identify a drought again later in the dry season in spring. As such, the original combined method therefore identifies two droughts (i.e. one in summer during the NAM and one in spring after a prolonged period without flow, see also Section 4.1). As indicated in the introduction, flow events replenish rootzone moisture below the channel bed, which is used by the riparian zone vegetation during the dry season for transpiration. Therefore, the occurrence of a drought or wet spell during the positive flow season, directly impacts moisture availability afterwards. As such, it is important that a given

hydrological state continuous from the wet into the dry season. In order for a hydrological drought to prolong from the wet into the dry season, drought should also be identified at lower consecutive drought/dry day numbers, which currently is impossible.

Third, originally the method was not developed to identify wet spells for zero flow conditions, as these would always occur during situations of low consecutive drought/dry day number. For the San Pedro, this would result in a wet spell to occur immediately at the start of the dry season, even if a drought was observed during the positive flow season (see also Section 4.1)."

We also extended the explanation of the new combined procedure presented, following an approach similar to the one presented in the original work of Van Huijgevoort et al (2012) making use of both mathematical symbols and enumerated lists. Here we use a figure to indicate the different steps.

Furthermore, in line with suggestion raised by the reviewer, we moved the figure that showed the difference between the old and new combined method to a newly created Section 4.1, which contains the following:

"4.1 Comparison of the original and modified combined method
Figure 5 shows the difference in identified discharge percentile between the combined method as proposed by Van Huijgevoort et al. (2012) (old method) and the modified combined method proposed here (new method). As indicated in Section 3.2 the old method was unable to identify drought during zero flow conditions with a low consecutive drought/dry day number. This can clearly be observed from Fig. 5b where these days, that generally occur at the beginning of the dry season during the fall, show high corresponding discharge percentiles. Furthermore, for the years 2004–2005 and 2009–2010, the drought observed during the positive flow season, first ceases and return the following spring at the end of the dry season for high consecutive drought/dry day number. This situation does not occur for the new method presented here, where the consecutive drought/dry day number of a given zero flow day is compared with its number observed during other years. As such, discharge percentiles in the fall can be small enough to indicate a drought. As a result, for both 2004–2005 and 2009–2010 the drought continued from the wet season into the fall, as an indication of the moisture state of the riparian zone. The fact that the observed hydrological drought continues after precipitation ceases results in a considerable increase in the total number of drought days for 2002–2005 and 2009–2011 (Fig. 5c).

Section 3.2 also mentioned that, the original method of Van Huijgevoort et al. (2012) would lead to high discharge percentiles at low consecutive drought/dry day number and as such was not specifically developed for this. For the fall, Fig. 5b shows for station 11 this would indicate the occurrence of a wet spell during the fall of almost each year, irrespective of the strength of the NAM. The newly combined method presented here, does not show this strong dependency. In stead, high discharge percentiles in the fall only occur during years where already during the NAM a wet spell was observed (i.e. 2000–2001 and 2006–2007). During the other years discharge percentiles in the fall do not indicate the occurrence of a wet spell."

We believe these changes have improved the description of the new combined procedure presented here and its applicability to identify hydrological anomalies within semi-arid ephemeral channels.

**Major concerning 3:**
As the current work focuses on hydrological anomalies within semi-arid environments and more specifically the San Pedro basin, we feel that only focusing on the ephemeral channels as suggested by reviewer 2 would not provide a complete picture of hydrological anomalies occurring throughout this region. Instead we assess their occurrence of three typical cases: 1) dry lands channel beds that only show runoff from intense precipitation events during the NAM and from remnants from hurricanes in the fall (NB), 2) the upland regions that have perennially flow conditions (BU), and 3) low land regions with perennially flowing conditions received from baseflow upslope (BD). To introduce this we added the following to the introduction:

"Results obtained for ephemeral channels as derived using the new combined procedure presented here, will be compared with observations from perennial rivers at higher elevation

as well as for downslope locations receiving continuous flow from upslope. By analyzing the occurrence of drought and wet spell anomalies across these locations, this work will provide a detailed overview of their occurrence and variability within the San Pedro, as well as the role of climate and local geographical location."

For case a) dry lands channels, the occurrence of a single event can generate a wet spell even after runoff ceases. This might be counterintuitive, as indicated by all three reviewers, as "How can zero flow conditions correspond to a wet spell". We hope that the comments provided by major point 1 and 2 as given above, have resolved this aspect.

Reviewer 2 indicated that one option would be to only focus on case a). As we did a poor job describing the different system, we understand this suggestion. However, we hope by correctly adding these details throughout the paper, it becomes clear why it is important also to address the uplands.

For case b) the hydrological response of these the upslope regions is much more similar to temperate environment. Hydrological drought and wet spells have a direct link to the amount of water available in the river network. Furthermore, as flow conditions are always positive, we added the following to Section 3.2:

"for locations within continuously flowing condition (BU and BD category) the TLM approach was used solely."

For case c), at shorter time scales (30-day moving average window), observed hydrological drought and wet spell characteristics resemble those observed for the upland reaches, since during the dry season all water originates from these uplands. However, also in these environments, transmission losses form an important source of moisture for the surrounding riparian zone. Since these channels transport the majority of their water during the NAM, when focusing on a longer timescale (e.g. MA-window of 1 year), their hydrological drought and wet spell characteristics resemble more case a) the NB category. As such, at these timescales a hydrological anomaly is more representative for the state of the riparian zone. Again, the paper did a poor job in explaining this.

Because of this difference in behavior across MA-window scales, we felt it was important to assess this. Therefore, we presented two figures for 30-day MA windows (Figs 6-8), various MA-window sizes (Fig. 10) and a one-year MA-window size (Figs. 9 and 11). However, we did not properly motivate these choices and have tried to include this in the updated version of the paper, by addressing these in both the Results section as well in within the Discussion. The latter now states:
"The mean duration of a wet spell last longer for upslope domains (BU) as compared to the lower elevation categories (BD and NB). For the upslope regions, shallow groundwater is expected to have a stronger control on the observed amount of baseflow for a longer period of time, increasing the mean duration (see also Section 4.2)."

Besides addressing these major concerns we have:

- **Changed the title as suggested by the reviewers**
- **Include more up to date references in introduction and section 3**
- **We propose to alter the setup of the manuscript by creating a new paragraph 4.1, before the original sections 4.1-4-3 which will become section 4.2-4.4.**
- **For the one-year MA-window analyses in Figures 11 and 13, we have added the details of why we feel this is interesting both in the Introduction as well as in Section 3.4 and in the Discussion. We will also highlight that this situation effectively corresponds to applying the TLM only.**
- **To decrease the length of the discussion and not to present new results we have moved the figures of the discussion of the original manuscript to a newly created section 4.5.**
- **Adjust the original figures where suggested.**

**We are convinced that by resolving these concerns and addressing the various issues raised by the reviewers the quality of the manuscript has improved consirably.**

The authors deal with a modification on the definition of hydrological drought that is supposed to be more robust for ephemeral rivers with prolonged zero-flow periods. While I understand the reasons behind the need of such modification of the van Huijgevoort et al. method, I find the message not well communicated and the proposed solution not fully on point. Both the tackled issue and the proposed solution need, in my opinion, to be better described, maybe with the help of a visual representation of a real study case (for a specific event/year) or even an artificial case (that highlight the key drawback of the current method) before showing the performance on all the available data.

**We would like to thank reviewer 2 for the constructive comments. This really helped us to evaluate the quality of our manuscript an assess the current bottle necks. Besides making these major changes to the paper, we have tried to carefully address the major and minor comments as given by reviewer 2, which are presented below.**

At the current state, it is difficult to gasp how the suggested modification actually works, since the full description is based only on text, and the examples in Fig. 4 is really hard to read. Also, since the method is needed for zero-flow rivers, the authors should focus only on such stations rather than all the available stations. If I understand correctly, the authors state that the method of van Huijigevoort et al. may "break" a drought event in two events if an event start during the wet period and continue during the dry one, but this should be highlighted in a figure that show one of such case and how the method solve the issue.
My understanding is that the author define a cdf of number of (antecedent) dry days that is different for each day of the year, rather than the same for all the day based on the total length of the zero-flow. This seems a solution to avoid "breaks" for events that started during the wet period, but can lead to difficulties for events starting at the beginning of a dry period. If my understanding is correct, I suggest to the authors to consider, first of all, if their goal is to produce an indicator that can be update in near real time (while the event is developing) or that defines the drought on past data. In the second case, better solutions can be found than the one proposed. Since a "true" definition of drought/wet spell start and length is not available (for obvious reasons), the authors need to clearly highlight that their outcomes are at least more reasonable that the one obtained with the previous method and not just as arbitrary.

**We agree with the reviewer that the original manuscript was not very clear in explaining the different steps and indicating the benefit of the newly presented approach. This was partly due to the fact that it was our interest to compare the different flow domains throughout the region, as indicate our general response above. As a result we did not want to put too much emphasis on the newly presented approach, but did feel it was important present these changes wrt the original method. As indicated in major concern 2 given above, we have updated the manuscript and better describe the newly presented approach. Furthermore, in line we major concern 1, we now motivate why we want a drought that starts in during the NAM to continue during the no-flow season afterwards. We have updated figure 5 of the manuscript which now shows the year 1973. This figure shows that for new approach, this "catching up" is not occurring anymore. We agree with the reviewer that the original manuscript did not emphasize this aspect. Therefore, we have decided to stress this aspect more by adding additional text in Section 3.2 of the manuscript as well as within a newly created section 4.1.**

[Figure]

A second point of contention for me is the need for a better explanation on the reasoning behind this kind of definition of hydrological drought and (especially) wet spells in zero-flow rivers. As the authors stated, they are looking at an issue that arise for specific rivers, with zero-flow during most of the year and flow only during monsoon, but it is not clear what is the goal of having hydrological drought (or wet spells) defined for such rivers in such a way. While the classic analysis of dry spells (length of periods of zero-flow) or wet spells (length of period with positive flow) in such rivers is relevant, I do not understand, for instance, the reasoning to define a day as part of a "wet spell" even if the flow is zero (e.g., first half of 2007 in Fig. 6a).

**We agree that the original version of the manuscript did not do a good job in indicating why we chose to focus on the three different flow domains, and why we opted to focus on drought and wet spells for rivers with zero-flow conditions occurring throughout the year. We would like to refer to our response to major concern 1. We believe the updated version of the manuscript better addresses this issue, as it was clearly lacking in the original work.**

Finally, the title of the paper is ambiguous, since the focus in on a redefinition of drought events in zero-flow rivers but the title seems to imply that the effects of elevation and flow dynamics in semi-arid rivers will be discussed.

**The original manuscript was indeed not very clear on why the focus was on elevation and flow dynamics. We believe that by adding this information, as stated in the previous point, the focus of the paper on both rivers with and without baseflow within eastern Arizona becomes**

**clear. We will update the title accordingly. Furthermore, we will thoroughly address the major concerns identified above. By doing so, we believe the paper becomes less ambiguous.**

Follow some minor comments:
P2L30. Why wet spells on rivers are defined based on precipitation here?

**For southern Arizona we believe that precipitation is the main driver of hydrological drought. At shorter timescales, groundwater releases can cause a wet spell but there impact is only visible for the perennially flowing rivers. We have stated this in the introduction.**

P2L39. This is not true for all the cases. There are plenty of evidences that over some regions less extreme drought are expected (e.g., northern Europe).

**The reviewer is indeed correct.**

P2L50. What is the relevance for water managers in rivers with zero-flow? Are those rivers under any water managing?

**We refer to our response to major concern 1. These anomalies are not directly relevant for water managers, but instead indicate the rootzone moisture availability within the riparian zone.**

P4L88. Why the river is here identified as perennial, whereas is defined ephemeral in the rest of the text (see e.g., P3L74) or just partially perennial (P4L103). Please be consistent.

**We have changed this into:**
**"As flow conditions at lower elevations for the majority of channels throughout this domain are ephemeral,"**

P4L120. You should focus only on the NB rivers, and eventually show that your method works for the other rivers too (if this is the case). P6L142. This should read as: "The TLM has a problem for locations with zero flow (in a specific period) for a considerable amount of years". Please reword.
**We refer to our major concern 1. The original manuscript did a poor job in describing why we wanted to differentiate between these three domains and not just present an updated version of the paper by Van Huijgevoort et al. (2012).**

P7L181. It is really difficult to extrapolate how the two methods work from fig. 4b. If 2003-2004 is a good example year, please make a specific figure that highlight the key differences between the two methods.

**We have added Fig. 5 (see above) to highlight this aspect and are currently creating a schematization figure where both methods are compared.**

P8. Sensitivity Analysis. This section does not seem well thought-out in my opinion. The range of values adopted in this analysis need to be better supported by some reasoning (e.g., pooling up to 180 days? moving windows of 5 years?).

**W.r.t to the length of te MA windows, we refer to our general response given above. Concerning the pooling, we agree with the reviewer that pooling up to 180 days is extreme. The reason why we decided to do this is to show that results do not very much, with these long pooling windows. However, we did not address this properly in the first submission of the manuscript. We will ensure that this will be treated in an updated version of the discussion (as also raised by the reviewer).**

P9L265. This description of the behavior of drought is a consequence of your definition of theevents rather a fact. You need "independent" evidences on the behavior of drought to support that your reconstruction is more adherent to the reality than the one obtained with the previous method.

**Correct. However, we feel that for semiarid regions the method presented here is able to represent the expected behavior as indicated above. However, we agree that for other locations this does not hold.**

P10L290-300. This analysis on the long-term variations is out of topic and not well supported by formal trend tests and analyses.

**We do not agree as we believe there is scientific literature to motivate these assumptions. Without accounting for long-term variations, the San Pedro would show a continuous drought starting in the 1990's which is solely the result of human behavior. Although this is technically correct, we believe that it does not provide the real message concerning the observed hydrological state. However, this was not clearly defined in the original submission. We added the following statement to Section 3.3:**

**"Although, the occurrence of these type of changes was rare for the San Pedro, it was observed for a few locations including station 3 (see also Fig. 2), where both a gradual decrease and shift in the late fifties was observed (see also Fig. 2). By not identifying and correcting for trend and step changes, results would show a drying out of the system, with a wet spell being the predominant condition during the first twenty years, and a drought being dominant during the last twenty years. Although, technically this is correct, we believe this would only reflect the impact of human behavior with increased amount in water uptake throughout time and added modifications throughout the catchment. As stated before, it is the interest of the current work to understand the behavior of hydrological anomalies and how these vary between locations. To be able to assess these, the identification and correction of shifts and trends is necessary, which subsequently allows for the identification and analyses of anomalies within the hydrological system (e.g. Villarini and Smith, 2010; Sadri et al., 2016)."**

P11. Fig. 8. This figure is rather confusion, and, in my opinion, not the best way to convey the key finding that the authors want to show here.
**In line with the suggestion raised by reviewer 1, we have changed this figure into 6 separate panels. We believe that this improve the readability, shows the dominant behavior of the North American Monsoon and enables differentiation between the three different flow domains representative for eastern Arizona.**

P12. Section 4.3 is this on only one river station or all the stations combined? This is not clear.
**This is for all stations combined and has been added to the text.**

P13L374. If a backward moving window is used (rather than a most common centered one) this need to be clarified and justified in the methodology.
**This is indeed an important point raised by the reviewer! Given that the hydrological state shown here is indicative for the state of the channel bed including vegetation, a backward looking window is justifyable. We include this statement in Section 3.4.**